# Solid Phase-Based Microextraction Techniques in Therapeutic Drug Monitoring

**DOI:** 10.3390/pharmaceutics15041055

**Published:** 2023-03-24

**Authors:** Sofia Soares, Tiago Rosado, Mário Barroso, Eugenia Gallardo

**Affiliations:** 1Centro de Investigação em Ciências da Saúde, Faculdade de Ciências da Saúde, Universidade da Beira Interior (CICS-UBI), 6200-506 Covilhã, Portugal; 2Laboratório de Fármaco-Toxicologia, Ubimedical, Universidade da Beira Interior, 6200-284 Covilhã, Portugal; 3Serviço de Química e Toxicologia Forenses, Instituto de Medicina Legal e Ciências Forenses—Delegação do Sul, 1169-201 Lisboa, Portugal

**Keywords:** sample pretreatment, solid phase-based microextraction techniques, human biological specimens, therapeutic drug monitoring

## Abstract

Therapeutic drug monitoring is an established practice for a small group of drugs, particularly those presenting narrow therapeutic windows, for which there is a direct relationship between concentration and pharmacological effects at the site of action. Drug concentrations in biological fluids are used, in addition to other clinical observation measures, to assess the patient’s status, since they are the support for therapy individualization and allow assessing adherence to therapy. Monitoring these drug classes is of great importance, as it minimizes the risk of medical interactions, as well as toxic effects. In addition, the quantification of these drugs through routine toxicological tests and the development of new monitoring methodologies are extremely relevant for public health and for the well-being of the patient, and it has implications in clinical and forensic situations. In this sense, the use of new extraction procedures that employ smaller volumes of sample and organic solvents, therefore considered miniaturized and green techniques, is of great interest in this field. From these, the use of fabric-phase extractions seems appealing. Noteworthy is the fact that SPME, which was the first of these miniaturized approaches to be used in the early ‘90s, is still the most used solventless procedure, providing solid and sound results. The main goal of this paper is to perform a critical review of sample preparation techniques based on solid-phase microextraction for drug detection in therapeutic monitoring situations.

## 1. Introduction

Therapeutic drug monitoring (TDM) is an individualized and personalized approach to monitor the levels of drugs that have narrow therapeutic windows, marked pharmacokinetic variability or a critical threshold for pharmacological action. This concept relies on the control and/or adjustment of dosages in order to obtain systemic concentrations of the prescribed drug that are associated to therapeutic efficacy, aiming at reducing the risk of toxic and adverse effects, optimizing clinical results via maximizing successful outcomes; in addition, it allows for the evaluating of safety profiles and adherence to therapy [1,2,3]. Requests for TDM may be justified by drug interactions, confirmation of adherence to therapy, efficacy, toxicity prevention and therapy interruption monitoring [4]. In TDM, plasma is considered one of the gold-standard samples, and this approach is routine practice for therapies involving antipsychotics, antiepileptics, antidepressants, antiarrhythmics, antivirals/antiretrovirals, anticancer drugs, antibiotics, antifungal drugs and for some immunosuppressants, except for cyclosporine and tacrolimus, for which whole blood is used for TDM purposes [1,2,5].

In recent years, interest in developing new sample pretreatment procedures in laboratories or industry has been increasing, largely due to concerns about the environmental impact of chemicals. Developing ecological and environmentally friendly methodologies usually implies using techniques that employ smaller amounts of sample and organic solvents; these include green chemistry methods, such as miniaturized preparation techniques. Consequently, developing more selective, sensitive and energy-efficient analytical methods deserves a great deal of attention.

Microextraction procedures can be categorized into solid phase- and liquid phase-based techniques [6,7]. Procedures based on solid-phase microextraction, involving adsorption or absorption of the compounds of interest onto a solid sorbent or film, can be divided in microextraction by packed sorbent (MEPS), solid-phase microextraction (SPME), micro solid-phase extraction (μ-SPE), dispersive solid-phase extraction (DSPE) or matrix solid-phase dispersion, magnetic solid-phase extraction (MSPE) and molecularly imprinted polymers (MIPs). These represent the most widely used techniques for drug determinations in TDM. Still, other approaches may be used for the same purpose, namely in tube-SPME, pipette tip solid-phase extraction (pipette tip SPE), solid-phase dynamic extraction (SPDE), thin film microextraction (TFME), stir bar sorptive extraction (SBSE) and fabric phase sorptive extraction (FPSE).

This review will address the use of miniaturized procedures based on solid-phase microextraction in sample preparation for TDM purposes. A brief description of the characteristics and functioning of each of these approaches will be presented and critically discussed, focusing on studies developed for drug analysis. No review dedicated to solid-phase microextraction-based methods for TDM has been published so far. The rate of consumption of these classes of compounds is relatively high, potentially impacting public health, and therefore it is important to compile the existing literature on the matter to assist health professionals in improving the quality of treatment for patients.

## 2. Methods

The systematic literature search was performed using the PubMed and ISI Web of Knowledge databases. The search strings were “therapeutic drug monitoring” or “TDM” combined with the different types of solid-phase microextraction-based pretreatment of human biological samples: “microextraction by packed sorbent” or “MEPS”, “solid-phase microextraction” or “SPME”, “in tube solid-phase microextraction” or “in-tube SPME”, “pipette tip solid-phase extraction” or “pipette tip SPE”, “micro solid-phase extraction” or “micro SPE”, “dispersive solid-phase extraction” or “DSPE”, “matrix solid-phase dispersion” or “MSPD”, “magnetic solid-phase extraction” or “MSPE”, “solid-phase dynamic extraction” or “SPDE”, “molecularly imprinted polymers” or “MIPs”, “thin film microextraction” or “TFME”, “stir bar sorptive extraction” or “SBSE” and “fabric phase sorptive extraction” or “FPSE”. Only papers with application to real samples were considered for analysis. Concerning molecularly imprinted polymers, only the last two years were included due to the high number of papers available. A similar situation occurred with solid-phase microextraction, for which results from the past three years were considered. In the case of microextraction by packed sorbent, dispersive solid-phase extraction or matrix solid-phase dispersion and magnetic solid-phase extraction, papers back to 2017 were evaluated, while for micro solid-phase extraction and fabric phase sorptive extraction, 2015 was used as starting year for analysis. For the remaining techniques, papers since 2010 were evaluated, due to the low number of publications available. All articles were independently selected by three of the authors in order to determine their importance in the context of the current review, and only those that were chosen by at least two of the authors were included.

## 3. Sample Pretreatment—Solid Phase-Based Microextraction Techniques

Taking into account that the preparation of biological specimens is the most time-consuming and laborious step for laboratories, the authors intend with this paper to perform a comprehensive review on the most recent techniques based on solid-phase microextraction for sample pretreatment in TDM. These miniaturized approaches were selected due to their growing use and because of the increased interest in the principles of green chemistry, leading to a lesser use of organic solvents and samples. Figure 1 summarizes all solid phase-based microextraction approaches for biological sample preparation in TDM.

In the following sections, and for all the above-mentioned approaches, each technique will be briefly described, as well as its applications in TDM.

### 3.1. Microextraction by Packed Sorbent

The MEPS procedure is a miniaturized version of the classic SPE technique, which was developed with the aim of reducing sample volume, organic extraction solvents and pretreatment time, without compromising extraction efficiency and allowing direct injection into gas (GC) and liquid (LC) chromatographic systems, due to the possibility of automation without any device modification [8,9,10,11]. There are several sorbents used and commercially available for this technique, including silica matrices (unmodified silica, ethylsilane (C2), octylsilane (C8) and octadecylsilane (C18)), C18 strong cation/anion exchange (SCX, SAX), mixed sorbents (C8/SCX), polystyrene-divinylbenzene copolymers (PS/DVB), restricted access material (RAM) and MIPs [9,10]. The extraction of analytes from the C2, C8 and C18 sorbents occurs via adsorption/hydrophobic van der Waals interactions and by electrostatic attraction and charge–charge Coulomb interaction for SCX and SAX sorbents. The RAM phases simultaneously present size exclusion with hydrophobic and ion-exchange extraction interactions, and the hydrophobic structure of the PS-DVB interacts with the analytes via van der Waals forces and π-π interactions of the aromatic rings. Interactions for MIPs are similar to those between antibodies and antigens, consisting in electrostatic and hydrophobic interactions, hydrogen bonds and van der Waals forces [12,13,14]. This technique combines sample extraction, pre-concentration and cleaning in a single device, and it is considered more environmentally friendly compared to conventional approaches [8,9,10,15]. Normally, a MEPS protocol follows a four-step procedure: conditioning of the sorbent, sample loading, washing and analyte elution. These steps consist of cycles of ascending and descending movements of the solutions through the sorbent, and all the stages and solvents used can be optimized in order to obtain better efficiency and recovery results of the analytes under study. In addition, other conditions such as sorbent selection, pH of the working solution and ionic strength must be optimized to reduce possible interferences, matrix effects and carry-over [9,10]. Table 1 indicates some of the advantages and disadvantages of this technique [9,11].

MEPS has been applied for the extraction of many drugs and metabolites from biological specimens, and Table 2 summarizes the bioanalytical procedures published between 2017 and 2022 which use MEPS for TDM.

Regarding serum samples, Szultka-Mlynska and Buszewski [19] developed a method for the identification of five immunosuppressants (cyclosporine A, everolimus, mycophenolic acid, sirolimus and tacrolimus) in 150 µL of serum, with sample preparation accomplished by MEPS using C18 sorbent and analysis by liquid chromatography coupled to tandem mass spectrometry (LC-MS/MS). The method was linear over a range of 1 to 50 ng/mL for all compounds, and it was applied to post-transplant patients. More recently, Cruz et al. [20] established a method for the therapeutic monitoring of some antipsychotics (chlorpromazine, clozapine, olanzapine and quetiapine) in only 100 µL of plasma, with extraction by MEPS, analysis by ultra-high pressure liquid chromatography coupled to tandem mass spectrometry (UHPLC-MS/MS) and application to samples from schizophrenic patients. Using a restricted access carbon nanotube as a selective stationary phase for extraction, they reached a limit of quantification (LOQ) of 10 ng/mL for all compounds and concluded that the method is capable of determining these antipsychotics at sub-therapeutic levels in a reduced sample volume. Marasca et al. [21], using the MEPS technique with a C2 sorbent and analysis by liquid chromatography with sequential spectrophotometric and spectrofluorimetric detection, developed a methodology for monitoring eight antidepressants and metabolites (sertraline, norsertraline, fluoxetine, norfluoxetine, citalopram, N-desmethylcitalopram, N,N-desmethylcitalopram and vortioxetine) in 100 µL of whole blood and oral fluid. The method was validated, LOQs between 1 and 10 ng/mL were obtained, and it was applied to patients suffering from major depression and/or related disorders. Previously, Magalhães et al. [18] validated a methodology for two antidepressants and a metabolite in plasma (500 µL), using a C18 sorbent and analysis by liquid chromatography with fluorescence detection, for which an LOQ value as low as 20 ng/mL was obtained for fluoxetine and norfluoxetine.

It should be noted that the MEPS technique was used by Xiong et al., who developed two methods, one for the determination of catecholamines and metanephrines in urine [23] and another for the quantification of metanephrines in plasma [24], with the objective of being applied in routine clinical laboratories for the diagnosis and screening of pheochromocytoma and paraganglioma neuroendocrine tumors.

### 3.2. Solid-Phase Microextraction and in-Tube Solid-Phase Microextraction

SPME is a popular and innovative solventless extraction technique that is used, for instance, in the pharmaceutical, forensic and environmental areas. This technique combines sampling, extraction and injection of samples into an analytical system, and, although it is not an exhaustive process, all the analyte obtained from the extraction is introduced into chromatography [25]. This technique uses fused silica fibers covered by a stationary phase (liquid or solid) in a syringe that will be exposed to a sample; an equilibrium will then be established between the sample and the sorbent, operated either by direct extraction, headspace extraction or membrane-protected extraction [25,26]. There are several coatings used for this technique, including polydimethylsiloxane (PDMS), polyacrylate (PA), divinylbenzene (DVB), PDMS/DVB, carbowax/DVB, PDMS/carboxen (CAR), carbowax/templated resin, DVB/CAR/PDMS, MIPs and RAM. SPME fibers can be categorized into two main groups, adsorbents or absorbents, depending on the type of collection process; the examples given above represent the first case, presenting van der Waals or hydrogen bonds interactions. The large implementation of SPME in laboratories leads to the creation of different types of fiber for different uses of this extraction technique, which is applicable to solid, liquid and gaseous samples, allowing for lower solvent consumption, simplification of sample preparation and a shorter extraction time [25,26,27,28]. In contrast to the above-mentioned, in-tube SPME uses a capillary tube, of <1 mm internal section, usually coated with fused silica. However, fiber-packed, sorbent-packed and rod-type monolith capillaries are available as well [29]. In order to improve efficiency and specificity, the analytes in this technique are absorbed or adsorbed on the outer surface of the packing material. Able to operate as a flow-through or as a draw/eject extraction system, the analysis can be performed online or offline by LC or GC, after desorption of the extracted compounds via a mobile phase flow or a static desorption solvent. This technique can also be automated [28,30,31].

Table 3 summarizes some of the advantages and disadvantages of these two techniques [25,26,32].

Table 4 contains bioanalytical procedures published between 2020 and 2022 (SPME), and published between 2010 and 2022 (in tube-SPME) concerning TDM.

SPME and in-tube SPME have been applied for the extraction of many drugs in different biological samples. Grecco et al. [35] developed a methodology for the determination of drugs for the treatment of Parkinson’s disease in plasma samples using in tube-SPME and hydrophilic interaction chromatography coupled with tandem mass spectrometry. With only 200 µL of sample, the five drugs showed LOQ values between 1.2 and 170 ng/mL, and the method was applied to authentic samples. Using SPME, Li et al. [36] established a method for the determination of five antipsychotics in blood and urine. With ultra-performance liquid chromatography coupled to tandem mass spectrometry (UPLC-MS/MS) analysis, the authors were able to determine the analytes at sub-therapeutic values with LOQs between 12.5 and 25 pg/mL, with the only disadvantage of the high volume of urine applied (10 mL). More recently, Nazdrajic et al. [39] developed a method for the determination of immunosuppressants such as tacrolimus, sirolimus, everolimus and cyclosporine A in 200 µL of blood samples, with extraction by matrix-compatible solid-phase microextraction fibers (BioSPME) and analysis by microfluidic open interface coupled with tandem mass spectrometry. Comparing with the work mentioned in the previous section by Szultka-Mlynska and Buszewski [19], and although with values of limits of detection (LOD) and LOQs not equally low, these authors also validated the method with linearity between 1 and 50 ng/mL for tacrolimus, sirolimus and everolimus, and applied the methodology to samples from patients undergoing immunosuppression therapy.

Jing et al. [41] developed a method for the biomonitoring of four metabolites of organophosphate flame retardants, for screening them to assess exposure risk for humans and for their potential application in epidemiological studies on individual exposures in large populations. Roy et al. [42] established a method for determining the plasma protein binding of drugs in plasma samples for atenolol, morphine, acetaminophen, lorazepam, carbamazepine, diazepam and buprenorphine. Applying BioSPME (C18) as the extraction technique, the authors believe that this measurement is essential during drug development, for pharmacokinetic and pharmacodynamic studies and in clinical practice since it is important for TDM and personalized medicine. Although without validation data, Bojko et al. [43] developed a methodology for the detection of doxorubicin in vivo lung perfusion with extraction by BioSPME (C8+benzenesulfonic acid particles) for targeting residual micrometastatic disease, demonstrating the potential of the technique for real-time monitoring of the administered chemotherapy and the usefulness in adapting the strategy to personalized treatment.

### 3.3. Pipette Tip Solid-Phase Extraction, Micro Solid-Phase Extraction and Dispersive Solid-Phase Extraction or Matrix Solid-Phase Dispersion

Pipette tip SPE is routinely used in several areas. All sample manipulations are performed by aspiration and disposal through a pipette tip using a micropipette. Of the commercially available materials, the most used are mixed-mode cation exchange and C18 sorbents. This technique is simpler, fast, uses disposable materials and, given the small volume of the bed and the mass of the sorbent inside the tip, aims to minimize the volumes of specimen and organic solvents required for the conditioning and elution steps, accelerating the evaporation step process and providing higher yield and lower costs [28,44,45].

Micro-SPE is a miniaturized technique that uses a device comprising a porous membrane envelope that contains a small amount of sorbent. There are several commercially available sorbents such as activated alumina, C2, C8, C18, activated carbon (CA) that provides π-π and hydrophobic interactions, Haye-Sep B and A, Porapak R, carbograph (GC) and multi-walled carbon nanotubes (MWCNTs) also with π-π interactions; however, more recently, the most used are zeolite, silica, MIPs and metal-organic frameworks (MOFs) with electrostatic and hydrogen bonds interactions [46,47]. This methodology aims to reduce the dimensions of the device with the use of micro or nano materials, reduce the volume of sorbents and their operating time and also the consumption of organic solvents. All this, combined with the simplicity of processing, its enrichment factor, high selectivity and sensitivity, compatibility with separation and detection systems and with headspace and immersion modes, lower time and associated costs, justify all the benefits and advantages of the technique. However, this procedure also presents disadvantages such as fragility of the fibers, restricted range of stationary phases and carry-over [46,47,48].

DSPE is an alternative technique applied to viscous, solid and semi-solid materials. Usually, the samples are mixed with the sorbent, and the mixture is transferred to an extraction column to be packed; subsequently, this is where the washing and elution steps will be carried out, in order to extract and isolate the analytes from the sample. This technique does not require repeated centrifugation, filtration or extraction steps; the eluents can be analyzed by GC or LC, but it cannot be completely automated. The extraction step is omitted, leading to the use of fewer organic solvents and less sample preparation time, being a more flexible and robust technique for which there is no degradation or denaturation when applied at room temperature and mild atmospheric pressure. The effectiveness and selectivity will depend on the solid support used and the elution solvent chosen. The most used materials are reversed-phases, such as C18- and C8-silica bonded phases, normal-phase materials such as alumina and florisil with adsorption/polar interactions, silica that exhibits hydrophilic interaction with the solute based on charge-based interactions, hydrogen bonding, π–π and dipole-dipole interactions, carbon-based materials and MIPs [13,49,50,51].

Table 5 compiles some of the advantages and disadvantages of the aforementioned approaches [32,47,48,51].

Table 6 displays the bioanalytical procedures published between 2010 and 2022 (pipette tip SPE), between 2015 and 2022 (µ-SPE) and between 2017 and 2022 (DSPE or MSPD) concerning TDM.

Looking at Table 6, there are some applications of pipette tip SPE, µ-SPE and DSPE techniques used for pretreatment of biological samples, mainly plasma. In 2019, Koller et al. [55] developed a method for the determination of antipsychotics and metabolites in only 200 µL of plasma samples, with extraction by µ-SPE and analysis by LC-MS/MS. As one of the validation parameters, LOQ values between 0.18 and 1 ng/mL were obtained, and it was concluded that this method could be successfully implemented in the clinical laboratory and applied for routine TDM. The same working group developed and validated a methodology for the monitoring of eleven tyrosine kinase inhibitors in plasma samples. Additionally, with analysis by LC-MS/MS and extraction by µ-SPE, LOQs between 0.3 and 5 ng/mL were obtained using 300 µL of sample, and the authors concluded that the approach used in their clinical practice applied to TDM, contributed to the individualization of dose adjustment and the managing of adverse effects in patients with chronic myeloid leukemia [57]. Huang et al. [56] developed a method for the identification and quantification of eighteen multi-class antibiotics in 500 µL urine samples. The extraction was performed by QuEChERS DSPE and analysis by LC-MS/MS, for which LOQ values between approximately 0.4 and 105 µg/mL were obtained. Due to the LOD and LOQ values below the clinical dosage of most antibiotics, the authors concluded that this methodology could be applied in the detection of this class of drugs for preventing disease at sub-therapeutic levels and for monitoring programs of targeted compounds at trace levels.

Also noteworthy is the work carried out by Pinto et al. [59] for the determination of a group of drugs that included antipsychotics, antidepressants, anticonvulsants and anxiolytics, in 200 µL plasma samples. Extraction was performed by disposable pipette extraction (C18-BSA), which represents a simple and effective technique based on SPE. With LC-MS/MS analysis, they obtained LOQ values between 0.5 and 20 ng/mL, applied the methodology to samples of schizophrenic patients undergoing multidrug therapy and concluded that TDM could help in the therapeutic response and in dosage regimens, to avoid excessively high and potentially toxic drug concentrations, as well as in the monitoring of adherence to treatment [60]. Zhang et al. [61] developed a method for the determination of seven androgens and 17-hydroxyprogesterone in serum samples, with the objective of being applied in medical laboratories to patients with polycystic ovarian syndrome.

### 3.4. Magnetic Solid-Phase Extraction and Solid-Phase Dynamic Extraction

MSPE consists of a magnetic material dispersed in a sample in solution, easily recovered with the application of a magnetic field. For this methodology, magnetic nanoparticles are incubated in a liquid sample, whether unprocessed, diluted liquid or extract, for a certain period of time that will be optimized, with the purpose of adsorbing the analytes in the material and subsequent equilibrium. The magnetic nanoparticles can be recovered by applying a magnet or, less commonly, by centrifugation or filtration; they will be washed to remove weakly bound species, and the analytes will be eluted. The analytes are desorbed from the magnetic nanoparticles via a solvent or mixture of solvents, which will be recovered and further processed [62,63]. There are several magnetic materials implemented in this technique, such as polymers, carbon nanotubes, graphene composites, ionic liquids, deep eutectic solvents with hydrogen bond interactions, MOFs, boronate affinity materials, host–guest molecular recognition by supramolecules, aptamers, polydopamine (PDA), MIPs and MIP-carbon compounds and covalent organic structures (COFs), which present hydrogen bonds, the effect of pore size and hydrophobic and π-π interactions as the main adsorption mechanisms [64,65,66,67,68,69]. The non-covalent methods such as π–π stacking interactions are frequently used to prepare graphene-based composites [70], and ionic liquids provide hydrogen bonding, dipole–dipole and ionic interactions with the analytes [13]. The principle of molecular interactions of boronate affinity materials relies on the reversible covalent reaction between boronic acid ligands and *cis*-diol-containing compounds, and several secondary interactions including hydrophobic, ionic and hydrogen bonding can occur, depending on the structure of the boronate ligand and supporting material used [71]. Robust supramolecular systems are constituted by electrostatic, π–π, host–guest and hydrophobic–hydrophilic interactions, van der Waals forces and hydrogen bonds [72]. The binding forces that mediate aptamer–target interactions are hydrogen bonding, electrostatic interaction, hydrophobic effect, π–π stacking and van der Waals forces [73]. The most used analysis method is LC coupled to mass spectrometry (MS) or UV-Vis spectrometry [63]. MSPE is considered an ecological technique, requiring smaller amounts of sorbent material (which can normally be reused several times), shorter extraction time and a limited number of sample treatment steps [48,74].

MSPE has been applied for the extraction of some drugs and metabolites in biological samples, and Table 7 summarizes the bioanalytical procedures published between 2017 and 2022 that use MSPE for TDM.

Cai et al. [75] developed a method for the detection of five antidepressants and metabolites (venlafaxine, paroxetine, fluoxetine, norfluoxetine and sertraline) in plasma and urine samples of clinical origin. With UHPLC-MS/MS analysis, they obtained linearity in a range of 2.5 to 1000 ng/mL and a LOQ value between 0.51 and 2.46 ng/mL for all compounds under study. The authors concluded that the method showed great potential for TDM from clinical biological samples and that this monitoring was recommended, since antidepressant combinations increase the risk of drug interactions or overlapping toxicity. Li et al. [77,79] carried out two studies for the simultaneous determination of a large number of anti-tumor drugs in 100 µL plasma samples and analysis by UPLC-MS/MS. In both articles, the authors concluded that TDM is of great significance for the individualized treatment of cancer patients, and that the developed methodologies are suitable for TDM and pharmacokinetics studies. Also using only 100 µL of plasma, Qi et al. [78] developed a method for the determination of eight antimicrobials (linezolid, vancomycin, teicoplanin, tigecycline, imipenem, meropenem, voriconazole and micafungin), which they believe to be suitable for application in routine TDM in critically ill ICU patients. With UPLC-MS/MS analysis, the authors obtained LOQ values between 0.1 and 0.2 µg/mL and linearity intervals between 0.1 and 50 µg/mL. 

Kang et al. [81] developed a method for the determination of folic acid and riboflavin in urine samples. The methodology was applied to real samples, showing a promising application in the rapid analysis of free folic acid and riboflavin for clinical drug monitoring and treatment. Zhang et al. [82], on the other hand, established a methodology for the quantification of free testosterone and free androstenedione in serum samples, with the aim of improving the diagnosis accuracy of polycystic ovary syndrome in infertile women when combined with other clinical indicators.

Concerning SPDE, this technique uses stainless steel needles covered with a PDMS film and 10% of activated carbon and can be applied for the preparation of liquid and vapor samples. The procedure is performed by passing the headspace using a syringe through the tube, so that a fixed volume is pulled and pushed an optimized number of times, allowing the process to take place under dynamic conditions, keeping the headspace volume constant. The analytes under study are retained in the stationary phase and are subsequently desorbed by injection into a GC instrument. The capillary used is more robust than the fibers used in other extraction techniques, and it is not mechanically damaged. However, the analytes tend to be retained on the inner wall of the needle during the thermal desorption process, and the length of this coating can also be a problem for this same injection process [32,83]. For the SPDE procedure, between the years 2010 and 2022 and for the same research method in the databases described above, it was possible to find only one work, carried out by Rossbach et al. [84], for the biomonitoring of n-heptane and metabolites in blood samples. Using the headspace extraction technique and analysis by gas chromatography and mass spectrometry, they obtained LODs between 0.006 and 0.021 mg/L and concluded that these data could be helpful for the evaluation of a biological exposure limit of n-heptane in blood.

Table 8 summarizes some of the advantages and disadvantages of these two approaches [32,85,86].

### 3.5. Molecularly Imprinted Polymers and Thin Film Microextraction

The MIPs extraction device is a material developed by the polymerization of functional and cross-linked monomers around a template of a molecule, obtaining a highly cross-linked polymer with binding sites specific to the target analyte. As these synthetic polymers have specific recognition sites and, consequently, predetermined selectivity for a given analyte or structurally related group, the polymer is stable, robust, resistant and mimics the interactions of natural receptors to retain a target molecule. After analyte extraction and desorption, the sample and desorption solution are conducted through the monolithic capillary using a syringe infusion pump. Due to their larger surface area, MIPs provide higher extraction efficiency. This technique is used in several areas of research due to the easy preparation of the monolithic column and the low cost of the synthesis of the extraction material [32,48,87,88]. 

TFME combines sampling and sample preparation with various routine and on-site applications such as clinical and bioanalytical ones, and it is especially applied to hydrophobic and semi-volatile components with high distribution constants. This methodology implements an increase in the surface area that will lead to a greater volume of extractive phase, conferring an increase in mass absorption rates and an improvement in the sensitivity of the technique, which facilitates extraction kinetics and capacity [32,89]. TFME presents high versatility for the development of devices, and some examples of these materials are immunosorbents with selective antigen–antibody interactions, MOFs, aptamers, nanostructured sorbents, MIPs and ionic liquids [89]. Dip coating is the most used approach, where the support is dipped in a mixture containing an extractive phase that is dissolved or dispersed in an appropriate solvent or glue. Regarding instrument compatibility, the devices can be thermally desorbed (instruments with gas phase introduction) and solvent desorbed (analytical instrumentation compatible with liquid samples) [25,90]. The membrane can be attached to a support rod for better introduction into the analytical system, and after extraction, it can be wrapped around the rod and introduced into the injection system for analyte desorption [32].

Table 9 compiles some of the advantages and disadvantages of above techniques [48,87,89].

Table 10 contains the bioanalytical procedures published between 2021 and 2022 (MIPs) and published between 2010 and 2022 (TFME) concerning TDM.

MIPs and TFME have been applied for the extraction of drugs in several biological specimens such as plasma, serum and urine. Li et al. [91] developed a methodology for the determination of quetiapine and clozapine in 200 µL plasma and 500 µL urine, with extraction by TFME and analysis by high performance liquid chromatography coupled with ultraviolet detection. The authors obtained LOQ values of approximately 0.05 µg/mL for plasma samples and 0.01 µg/mL for urine samples, concluding that the method showed high potential as a powerful pretreatment technology for routine TDM in these biological samples. Shahhoseini et al. [93] established a method for the determination of seven tricyclic antidepressants and metabolites in 700 µL of plasma. With extraction by joining thin film molecularly imprinted polymer techniques and analysis by UHPLC-MS/MS, LOQ values between 1 and 5 ng/mL were obtained, and the method was applied to plasma samples from patients who were prescribed these antidepressants. More recently, Włodarski et al. [96] published a method for the determination of two β-Lactam antibiotics (imipenem and piperacillin) in plasma and bronchoalveolar lavage samples, using TFME for sample preparation. With LC-MS/MS analysis, and although without validation data presented for the bronchoalveolar lavage samples, they obtained an LOQ value of 0.01 mg/L for plasma samples. The authors concluded that the work demonstrates that only a small fraction of biologically active antibiotics reaches the site of infection, providing clinicians with a high-throughput tool for studies on personalized TDM when tailoring the dosing strategy to an individual patient.

Another possible application of the MIP extraction procedure is the work of Sardaremelli et al. [97] for monitoring hydrogen peroxide in plasma samples, fully validated and applied to unprocessed biological samples. Abdollahiyan et al. [98] have used MIPS for the identification of carcinoembryonic antigens in plasma samples, concluding that the developed extraction technique appears to be suitable for point-of-care applications in biomedical and clinical analysis.

### 3.6. Stir Bar Sorptive Extraction and Fabric Phase Sorptive Extraction

SBSE does not employ organic solvent, and it is a recognized method for pre-concentrating organic compounds on a coated stir bar, mainly from aqueous samples. It is a simple, robust and efficient methodology, with the possibility of automation, and can be applied to various matrices in diverse areas such as the clinical, food and environmental fields. The bar adsorbs the analytes during agitation, and this extraction technology is based on the equilibrium distribution between the analytes in the sample and the coating. Analytes are desorbed from the sorbent phase by thermal desorption compatible with GC and LC [48,99,100,101]. Coatings for stir bars are limited to non-polar polymers covering it, requiring combination with a derivatization step to include polar and thermally labile molecules, or the development of new materials [48,99]. The commercially available and most common coatings are PDMS, polyethylene (PEG) and PA; however, the applications of this technique may be expanded (making it more versatile) by using coatings such as poly(phthalazine ether sulfone ketone), polypropylene, PDMS/polypropylene, monolithic materials, nanocarbon materials, functional monomers, inorganic particles, MOFs, RAM and MIPs, which will increase selectivity and affect recovery and method dynamics [99,100,101].

FPSE is an evolutionary approach that deploys a natural or synthetic permeable and flexible fabric substrate to host a chemically coated sol-gel organic–inorganic hybrid sorbent, resulting in a versatile, fast and sensitive microextraction device. The membranes have high stability and allow the direct extraction of analytes without modifying the sample, minimizing the pretreatment steps. After membrane extraction, an organic solvent in small volume is used to back-extract the analytes and pre-concentrate them. Normally, solvent evaporation and sample reconstitution are not necessary. FPSE can utilize a variety of neutral, cation and mixed-mode cation exchangers, anion and mixed-mode anion exchangers, and zwitterionic and mixed-mode zwitterionic sorbents. The technique has been implemented for the extraction and determination of analytes at trace and ultra-trace concentrations in environmental and food samples, pharmaceutical products and biological samples [102,103,104].

Table 11 summarizes some of the advantages and disadvantages of these two approaches [48,100,101,104,105,106].

Considering the above, there are some analytical approaches to detect and quantify various substances applying SBSE and FPSE as extraction techniques for TDM. Table 12 summarizes the bioanalytical procedures published between 2010 and 2022 (SBSE) and between 2015 and 2022 (FPSE) concerning TDM.

Through the application of the SBSE extraction procedure, Catai et al. [108] developed a methodology for the determination of fluoxetine, sertraline, citalopram and paroxetine in 800 µL of plasma samples. With analysis by non-aqueous capillary electrophoresis with diode-array detection, they obtained LOQ values between 10 and 25 ng/mL, with linearity in the range of 20 to 500 ng/mL for fluoxetine. The results obtained demonstrated that the method is suitable for analysis of the antidepressants under study at therapeutic levels and for therapeutic monitoring purposes, but also in the evaluation of plasmatic levels in toxicological analysis after accidental or suicidal ingestion. On the other hand, Marques et al. [111] developed a method for the determination of fluoxetine in 240 µL of plasma with the same extraction technique and analysis by high-performance liquid chromatography with fluorimetric detection. The authors obtained an LOQ value of 32.67 ng/mL and linearity between 25 and 250 ng/mL and concluded that the method represented a potential tool since it allows results at concentrations compatible with TDM, with less endogenous interference and that it may help to understand the dose–response relationship of antidepressants in the human body. Kabir et al. [110] developed a method for the simultaneous monitoring of three drugs for the treatment of inflammatory bowel disease (ciprofloxacin, sulfasalazine and cortisone) in blood, plasma and urine samples, with extraction by FPSE and analysis by high-performance liquid chromatography with photo-diode array detection. LOQ values between 0.05 and 0.10 µg/mL were obtained for all biological samples, and the authors concluded that this new approach has a high potential as a fast, robust and green analytical tool for future clinical and pharmaceutical applications.

Another application for the FPSE extraction technique is the work carried out by Locatelli et al. [114] for the determination of six solar UV filters in blood, plasma and urine samples, for which the authors concluded that it opens a new direction in the analysis of these compounds with potential future applications in pharmacokinetics, pharmacodynamics and toxicology, with the objective of evaluating bioaccumulation and the adverse human health effects of personal care product ingredients.

## 4. Conclusions and Future Perspectives

The numerous classes of medications addressed in this review, such as antidepressants, antipsychotics, antiepileptics, immunosuppressants, anti-cancer drugs and antibiotics, are widely prescribed by health professionals for the treatment of various pathologies. The wide use of these drugs underlines the importance of developing methods to monitor their concentrations in biological samples from patients undergoing treatment with these medications. TDM will make it possible to assess the patient’s status and adherence to treatment and to individualize the therapy for each patient. This practice will help minimize the risk of interactions with other medications and possible toxic and secondary effects, increasing the probability of good results and enabling a better quality of life for patients. Several miniaturized procedures for sample treatment, mainly plasma, have been applied, and these include MEPS, SPME, µ-SPE, MSPE, MIPs, TFME and SBSE. From these, the most used are MIPs and SPME. All the studied techniques can be applied to TDM and to any class of these drugs, with the choice being made by the different sorbents available for the purpose. Among these techniques, the most applied in recent years have been MEPS, SPME, DSPE, MSPE and MIPs, and what is seen is a trend towards greater use of these miniaturized techniques to the detriment of extraction techniques considered classic, such as SPE. Despite being quite recent, extractions using fabric phases sound promising and appealing concerning sample and solvent reduction. The development of new methodologies for the detection and quantification of these compounds is based on the objective of using smaller volumes of biological specimens and organic solvents in the extracting phase, the implementation of simple and fast procedures, with little waste, and the capability of being automated. As prospects for the future, interest will remain in developing green techniques, as well as in what concerns fully automated miniaturized systems and new sorbents. The choice of robust analysis equipment, such as high resolution techniques, is also important due to the sensitivity and specificity that the methods allow and the detection of analytes of interest even when present at lower concentrations. Efforts should be now focused on improving the existing techniques, particularly concerning their main drawbacks, rather than on developing new ones. The main objective of the compilation of information regarding the developed methodologies is the availability of data for the improvement of the quality of the treatment of patients and, consequently, the improvement of the medical condition of each individual. In the future, it is expected that TDM will be performed with the routine implementation of these recent microextraction techniques. 

## Figures and Tables

**Figure 1 pharmaceutics-15-01055-f001:**
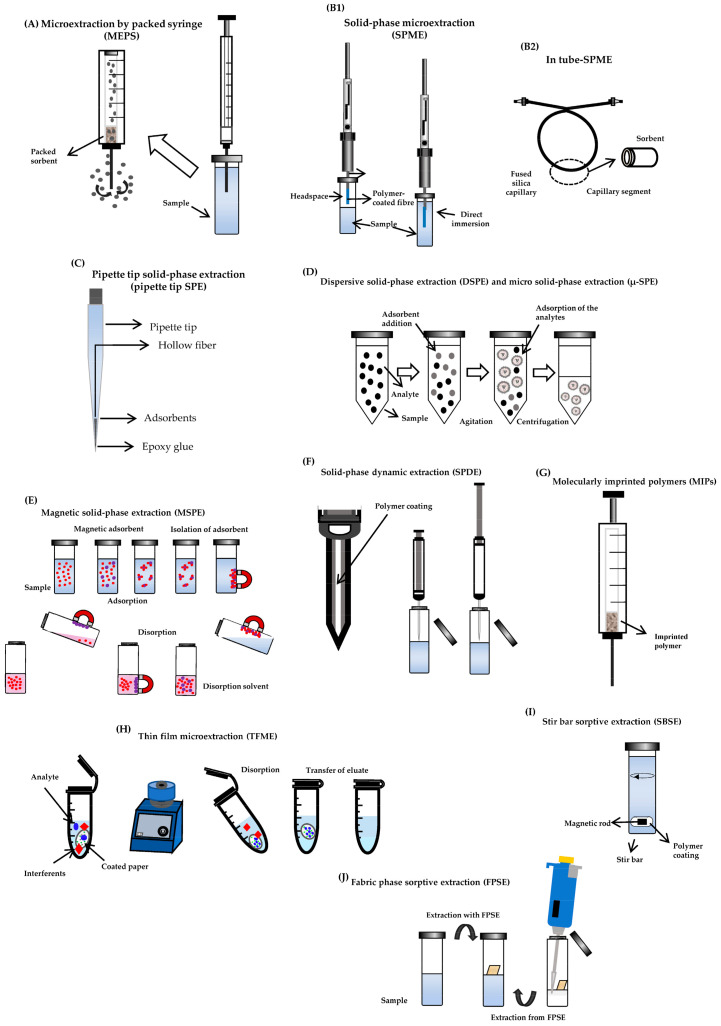
Solid phase-based microextraction approaches for sample preparation: (**A**) MEPS; (**B1**) SPME and (**B2**) in-tube SPME; (**C**) pipette tip SPE; (**D**) DSPE and µ-SPE (**E**) MSPE; (**F**) SPDE; (**G**) MIPs; (**H**) TFME; (**I**) SBSE and (**J**) FPSE.

**Table 1 pharmaceutics-15-01055-t001:** Advantages and disadvantages of the MEPS procedure.

Extraction Technique	Advantages	Disadvantages
MEPS	- Simple and broad (works with several ranges of analytes and matrices); - Environmentally friendly (uses low sample volumes and organic solvents); - Sorbents can be re-used several times, therefore the cost per analysis is lower; - Possibility of automation.	- Clogging of the sorbent; - Cannot process large sample volumes; - Carry-over.

**Table 2 pharmaceutics-15-01055-t002:** Bioanalytical procedures using MEPS approaches for TDM.

Analytes	Sample (Amount)	Sample Pretreatment and Extraction Procedure	Analytical Technique	LOD; LOQ	Linear Range	Ref.
Zonisamide	100 μL of plasma	Protein precipitation with 400 μL of ice-cold acetonitrile, agitation, centrifugation, evaporation and reconstitution with 100 μL of 0.1 M phosphate buffer solution (pH 8). MEPS (C18): conditioning with 3 × 200 μL of methanol and 3 × 200 μL of water; sample load (2 × 100 μL) at 10 μL/s; elution with 2 × 30 μL of acetonitrile and dilution with 90 μL of water.	HPLC-DAD	n.a. and 0.2 μg/mL	0.2–80 μg/mL	[16]
Lamotrigine	100 μL of plasma and saliva	Protein precipitation with 400 μL of ice-cold acetonitrile, agitation, centrifugation, evaporation and reconstitution with 200 μL of 0.3% triethylaminewater solution (pH 6). MEPS (C18): conditioning with 3 × 200 μL of methanol and 3 × 200 μL of water; sample load (3×) at 10 μL/s; washing with 200 μL of water; elution with 2 × 30 μL methanol and dilution with 90 μL of water.	HPLC-DAD	n.a. and 0.1 μg/mL for both samples	0.1–20 μg/mL for both samples	[17]
Fluoxetine, Norfluoxetine, Paroxetine	500 μL of plasma	Protein precipitation with 1.5 mL of acetonitrile, agitation, centrifugation, evaporation and reconstitution with 500 μL of 50 mM sodium phosphate monobasic anhydrous aqueous solution (pH 4). MEPS (C18): conditioning with 3 × 200 μL of methanol and 2 × 200 μL of water; sample load (3×) of the entire volume; washing with 2 × 200 μL of 5% aqueous ammonium hydroxide solution; elution with 5 × 200 µL of methanol with 1% formic acid.	HPLC-FLD	5 and 20 ng/mL (Fluoxetine, Norfluoxetine), 1 and 5 ng/mL (Paroxetine)	20–750 ng/mL (Fluoxetine, Norfluoxetine), 5–750 ng/mL (Paroxetine)	[18]
Cyclosporine A, Everolimus, Mycophenolic acid, Sirolimus, Tacrolimus	150 μL of serum	MEPS (C18): conditioning with 200 μL of methanol and 200 μL of water; sample load (3 × 150 μL) at 5 μL/s and air-dried; elution with 300 μL of methanol with formic acid.	LC-MS/MS (ESI)	0.021 and 0.063 ng/mL (Cyclosporine A), 0.023 and 0.068 ng/mL (Everolimus), 0.027 and 0.092 ng/mL (Mycophenolic acid), 0.029 and 0.098 ng/mL (Sirolimus), 0.031 and 0.113 ng/mL (Tacrolimus)	1–50 ng/mL for all the compounds	[19]
Chlorpromazine, Clozapine, Olanzapine, Quetiapine	100 μL of plasma	The sample is diluted with 400 μL of borate buffer solution (10 mmol/mL, pH 9). MEPS (RACNT): conditioning with 2 × 100 μL of acetonitrile and 2 × 100 μL of water; sample load (3×); washing with 150 μL of water; elution with 2 × 100 μL of acetonitrile.	UHPLC–MS/MS (ESI)	n.a. and 10 ng/mL for all the compounds	10–700 ng/mL (Chlorpromazine, Clozapine, Quetiapine), 10–200 ng/mL (Olanzapine)	[20]
Sertraline, Norsertraline, Fluoxetine, Norfluoxetine, Citalopram, N-desmethylcitalopram, N,N-desmethylcitalopram, Vortioxetine	100 μL of blood and oral fluid	VAMS (20 µL): absorption for 5 s, drying for 1 h at room temperature and ultrasound-assisted extraction for 20 min in 1 mL of methanol. Subsequently, for the solution resulting from the blood sample: evaporation, centrifugation and reconstitution with 100 μL of HPLC mobile phase (65:35, *v*:*v* of 33 mM aqueous phosphate buffer, pH 3 containing 0.3% triethylamine:acetonitrile) and for the solution resulting from the oral fluid sample: centrifugation. MEPS (C2): activation with 3 × 100 μL of methanol; conditioning with 3 × 100 μL of water; sample load (10×) at 5 μL/s; washing with 2 × 100 μL of water and 100 μL (10 mM, pH 9) of carbonate buffer:methanol (90:10, *v*:*v*) at 20 μL/s; elution with 3 × 200 μL of methanol at 5 μL/s.	HPLC-UV-FL	2.5 and 7 ng/mL (Sertraline, Norsertraline), 3 and 10 ng/mL (Fluoxetine, Norfluoxetine), 0.3 and 1 ng/mL (Citalopram, N-desmethylcitalopram, N,N-desmethylcitalopram), 1.5 and 5 ng/mL (Vortioxetine) for blood samples; 1.5 and 5 ng/mL (Sertraline, Norsertraline), 2.5 and 7 ng/mL (Fluoxetine, Norfluoxetine), 0.3 and 1 ng/mL (Citalopram, N-desmethylcitalopram, N,N-desmethylcitalopram), 1 and 3 ng/mL (Vortioxetine) for oral fluid samples	7–500 ng/mL (Sertraline, Norsertraline), 10–750 ng/mL (Fluoxetine, Norfluoxetine), 1–200 ng/mL (Citalopram, N-desmethylcitalopram, N,N-desmethylcitalopram), 5–500 ng/mL (Vortioxetine) for blood samples; 5–500 ng/mL (Sertraline, Norsertraline), 7–750 ng/mL (Fluoxetine, Norfluoxetine), 1–200 ng/mL (Citalopram, N-desmethylcitalopram, N,N-desmethylcitalopram), 3–500 ng/mL (Vortioxetine) for oral fluid samples	[21]
(R)- and (S)- Omeprazole	100 μL of plasma and oral fluid	The sample is diluted with water (1:4, *v*:*v*). MEPS (C8): conditioning with 100 μL of ethanol and 100 μL of water; sample load (6×); washing with 5% 2 × 100 µL of methanol in water; elution with 2 × 250 µL of ethanol.	LC-MS/MS (ESI)	0.1 and 0.4 ng/mL (calculated limits) for both samples and compounds	25–600 ng/mL (plasma) and 25–300 ng/mL (oral fluid) for both compounds	[22]

ESI: electrospray ionization; HPLC-DAD: high-performance liquid chromatography-diode array detection; HPLC-UV-FL: liquid chromatography-sequential spectrophotometric and spectrofluorimetric detection; HPLC-FLD: high-performance liquid chromatography-fluorescence detection; LC-MS/MS: liquid chromatography-tandem mass spectrometry; LOD: limit of detection; LOQ: limit of quantitation; MEPS: microextraction by packed sorbent; n.a.: not available; RACNT: restricted access carbon nanotubes; UHPLC-MS/MS: ultra-high liquid chromatography-tandem mass spectrometry; VAMS: volumetric absorptive microsampling. In the absence of the LOQ value, the lowest point of the calibration curve was considered.

**Table 3 pharmaceutics-15-01055-t003:** Advantages and disadvantages of the SPME and in-tube SPME procedures.

Extraction Technique	Advantages	Disadvantages
SPME and in-tube SPME	- Simple and solvent-free extraction method; - Reduces solvent consumption and time extraction since all the extracted material can be directly analyzed; - Extracting device is portable and allows field sampling; - Possibility of automation.	- Limitation on the chemical nature of the stationary phase on the market; - Competition between drug and endogenous compounds for the fiber; - Fibers are fragile materials, leading to breakage and coating stripping, limiting lifetime; - Non-exhaustive method.

**Table 4 pharmaceutics-15-01055-t004:** Bioanalytical procedures using SPME and in tube-SPME approaches for TDM.

Analytes	Sample (Amount)	Sample Pretreatment and Extraction Procedure	Analytical Technique	LOD; LOQ	Linear Range	Ref.
Rifampicin	500 µL of plasma	Protein precipitation with acetonitrile at a 2:1 ratio (*v*:*v*), agitation and centrifugation; the supernatant is collected, dried, resuspended with 0.5 mL of buffer solution and vortexed. In-tube SPME: extraction is performed with 10 draw/eject cycles; sample solution draw/eject volume of 200 µL; pH of the buffer solution of 7; draw/eject flow rate at 315 µL/min.	HPLC-UV	n.a. and 0.1 µg/mL	0.1–100 µg/mL	[33]
Interferon α_2a_	250 µL of plasma	The sample is diluted with 250 µL of phosphate buffer solution (0.025 mol/L, pH 6) and vortexed. In-tube SPME (immunoaffinity capillary): extraction is performed with 20 draw/eject cycles (150 µL), at a linear flow rate of 315 µL/min, using online desorption (dynamic desorption) by redirecting the mobile phase through the capillary.	HPLC-FD	n.a. and 0.006 MIU/mL	0.006–3 MIU/mL	[34]
Levodopa, Carbidopa, Benserazide, Dopamine, 3-O-methyldopa	200 µL of plasma	Protein precipitation with acetonitrile at a 1:2 ratio (*v*:*v*), agitation, centrifugation and filtration of the supernatant (400 μL). In-tube SPME (aminopropyl hybrid silica monolithic capillary containing mesoporous Santa Barbara Amorphous (SBA-15) particles)	HILIC-MS/MS (ESI)	n.a. and 22 ng/mL (Levodopa), n.a. and 33 ng/mL (Carbidopa), n.a. and 170 ng/mL (Benserazide), n.a. and 1.2 ng/mL (Dopamine), n.a. and 10 ng/mL (3-O-methyldopa)	22–2000 ng/mL (Levodopa), 33–2000 ng/mL (Carbidopa), 170–2000 ng/mL (Benserazide), 1.2–2000 ng/mL (Dopamine), 10–2000 ng/mL (3-O-methyldopa)	[35]
Perphenazine, Chlorpromazine, Chlorprothixene, Promethazine, Trifluoperazine	500 µL of blood and 10 mL of urine	Hollow fiber-based SPME: - Blood: the sample is mixed with 5 μL of ammonia water and vortexed for 5 s; the mixture is transferred into a tube equipped with a hollow fiber, the tube is cleaned 2 × 245 µL with water, and the cleaning solution is transferred into the tube; the device is conditioned in ultrasonic bath for 20 min; the hollow fiber is removed and placed into another tube containing 1 mL of methanol; the solution is ultrasonicated for 15 min to desorb the analytes and centrifuged. - Urine: the sample is mixed with 25 µL of ammonia water to adjust the pH to ~10 and vortexed for 5 s; the hollow fiber is immersed in the solution in the tube and ultrasonicated for 20 min; the hollow fiber is removed and placed into another tube with 1 mL of methanol and ultrasonicated for 15 min to desorb the analytes and centrifuged.	UPLC-MS/MS (ESI)	12.5 and 25 pg/mL (Perphenazine, Chlorprothixene, Promethazine), 6.25 and 12.5 pg/mL (Chlorpromazine, Trifluoperazine) for blood samples, 12.5 and 25 pg/mL (Perphenazine), 6.25 and 12.5 pg/mL (Chlorpromazine, Chlorprothixene, Promethazine, Trifluoperazine) for urine samples	25–1 × 10^4^ pg/mL (Perphenazine, Chlorprothixene, Promethazine), 12.5–1 × 10^4^ pg/mL (Chlorpromazine, Trifluoperazine) for blood samples, 25–1 × 10^4^ pg/mL (Perphenazine), 12.5–1 × 10^4^ pg/mL (Chlorpromazine, Chlorprothixene, Promethazine, Trifluoperazine) for urine samples	[36]
Tranexamic acid	n.a. of plasma and urine	SPME (hydrophilic-lipophilic balance (HLB) coated) - Plasma: preconditioning with 1.5 mL of a methanol:water (50:50, *v*:*v*) mixture for 10 min; the device is exposed to 1 mL of sample solution (sample/phosphate buffered saline (1:3, *v*:*v*)) for 5 min; the device is rinsed in a solution of 1 mL of water:methanol (90:10, *v*:*v*) for 10 s under static conditions; the device is desorbed in 1 mL of a methanol:acetonitrile:water (3:3:4, *v*:*v*:*v*) solution for 10 min. - Urine: preconditioning with 1.5 mL of a methanol:water (50:50, *v*:*v*) mixture for 10 min; the device is exposed to 1 mL of sample solution (sample/0.5 M phosphate buffered saline (1:3, *v*:*v*)) for 5 min; the device is rinsed in 1 mL of water for 10 s under agitated conditions; the device is desorbed in 1 mL of a water:methanol (90:10, *v*:*v*) solution for 10 min.	LC-MS/MS (n.a.)	10 μg/mL for plasma samples, 25 μg/mL for urine samples	10–1000 μg/mL for plasma samples, 25–1000 μg/mL for urine samples	[37]
Sorafenib, Dasatinib, Erlotinib hydrochloride	2 mL of plasma, serum and n.a. of urine	100 μL of hydrochloric acid (12 mol/L) and 100 μL of trifluoracetic acid is mixed with 2 mL of plasma which is agitated, centrifuged, and its supernatant separated and diluted with water (2:8, *v*:*v*), and the acid solution (pH 1) is neutralized with sodium hydroxide (0.01 mol/L) and filtered through a PVDF membrane. An amount equal to 2 mL of acetonitrile is added to 2 mL of serum which is centrifuged and its supernatant separated, filtered and diluted with water (2:8, *v*:*v*) before extraction. The urine sample is centrifuged, filtered and diluted with water (5:5, *v*:*v*) before extraction. TF-SPME (polyfam/Co-MOF-74 composite nanofibers): the piece of sorbent (1 cm^2^) is cut from the nanofiber sheet and submerged in 10 mL of acetonitrile for 10 min for conditioning; it is immersed in 20 mL of the sample solution (optimum pH 10) for adsorption under agitation for 10 min; the sorbent is transferred to a vial, to which 500 µL of alkaline methanol is added, plus stirring for 7 min for the desorption process.	HPLC-UV	0.03 and 0.1 μg/L (Sorafenib), 0.15 and 0.5 μg/L (Dasatinib), 0.2 and 0.5 μg/L (Erlotinib hydrochloride) for all the samples	0.1–1500 μg/L (Sorafenib), 0.5–1500 μg/L (Dasatinib, Erlotinib hydro-chloride) for all the samples	[38]
Tacrolimus, Sirolimus, Everolimus, Cyclosporine A	200 µL of blood	The sample is subjected to the mechanical lysis process of three freeze-thaw cycles (1 min in liquid nitrogen and 1 min in an ice bath); the sample is further subjected to an additional chemical lysis process with 1.3 mL of a lysing solution of zinc sulfate:acetonitrile:water (6:3:1, *v*:*v*:*v*). BioSPME (Oasis^®^ hydrophilic-lipophilic balance (HLB) particles): extraction for 60 min at 55 °C; the fiber is rinsed in water for 5 s; the fiber is placed into the pre-filled MOI chamber; the desorbed analytes are introduced into the equipment.	MOI-MS/MS (ESI)	0.3 and 0.8 ng/mL (Tacrolimus, Cyclosporine A), 0.2 and 0.7 ng/mL (Sirolimus), 0.3 and 1 ng/mL (Everolimus)	1–50 ng/mL (Tacrolimus, Sirolimus, Everolimus), 2.5–500 ng/mL (Cyclosporine A)	[39]
Valproic acid	50 µL of plasma	BioSPME (LC Tips C18): conditioning with 200 μL of a mixture of methanol:water (50:50, *v*:*v*) for 20 min under homogenization in an orbital shaker; 50 μL of the sample and 150 μL of hydrochloric acid 0.1 M is added in a polypropylene tube, followed by homogenization in an orbital shaker for 30 min; elution of the LC Tips C18 by adding the tips to a GC autosampler vial containing 150 μL of methanol, performing another homogenization step for 30 min.	GC-MS (n.a.)	n.a. and 10 mg/L	10–150 mg/L	[40]

BioSPME: matrix-compatible solid-phase microextraction fibers; ESI: electrospray ionization; GC-MS: gas chromatography-mass spectrometry; HILIC-MS/MS: hydrophilic interaction chromatography-tandem mass spectrometry; HPLC-FD: high-performance liquid chromatography-fluorescence detection; HPLC-UV: high-performance liquid chromatography-ultraviolet detection; in-tube SPME: in-tube solid-phase microextraction; LC-MS/MS: liquid chromatography-tandem mass spectrometry; LOD: limit of detection; LOQ: limit of quantitation; MOI-MS/MS: microfluidic open interface-tandem mass spectrometry; n.a.: not available; SPME: solid-phase microextraction; TF-SPME: thin film solid-phase microextraction; UPLC-MS/MS: ultra-performance liquid chromatography-tandem mass spectrometry. In the absence of the LOQ value, the lowest point of the calibration curve was considered.

**Table 5 pharmaceutics-15-01055-t005:** Advantages and disadvantages of the pipette tip SPE, µ-SPE and DSPE procedures.

Extraction Technique	Advantages	Disadvantages
pipette tip SPE	- Easy, fast and disposable materials are used; - Reduction in the volume of sample and solvent, and increased speed in the evaporation step; - Higher throughput and minimization of costs.	- Limited availability of commercial sorbents in tip format.
µ-SPE	- Reduction of the solvents’ operating time and chemical consumption; - Low solvent volume and time consumption; - High selectivity, sensitivity and simplicity of application; - Low cost and compatibility with various systems of analyte separation and detection.	- Fragile fibers; - Stationary phase with a restricted range; - Carry-over.
DSPE	- Simple, flexible and robust; - Reduction of solvent and time required; - Does not require repetitive centrifugation, filtration or extraction stages.	- Cannot be fully automated.

**Table 6 pharmaceutics-15-01055-t006:** Bioanalytical procedures using pipette tip SPE, µ-SPE and DSPE or MSPD approaches for TDM.

Analytes	Sample (Amount)	Sample Pretreatment and Extraction Procedure	Analytical Technique	LOD; LOQ	Linear Range	Ref.
Dextromethorphan	100 µL of plasma	300 μL of water and 50 μL of 1 mol/L glycine-sodium hydroxide buffer (pH 10) are added to the sample; the mixture is centrifuged, and the supernatant is reserved. Pipette tip SPE (MonoTip C18 tips (C18-bonded monolithic silica gel)): conditioning with 200 µL of methanol and 200 µL of water; extraction with 200 μL of the prepared supernatant performed for 20 sequential aspirating/dispensing cycles; washing with 200 µL of water; discarding the eluate; tip drying for 30 s; elution with 100 µL of methanol for 5 aspirating/dispensing cycles.	GC-MS (EI)	1.25 and 2.5 ng/mL	2.5–320 ng/mL	[52]
Verapamil	2 mL of plasma	Protein precipitation with 1 mL of acetonitrile, agitation and centrifugation. D-µSPE (Graphene oxide/polydopamine (PDA) and Graphene oxide/Fe_3_O_4_): 4 mg of GO/Fe_3_O_4_ and 2 mg of GO/PDA sorbents are dispersed into the sample and placed in an ultrasonic bath; drug-loaded GO/Fe_3_O_4_ is separated by applying an external magnetic field; GO/PDA sorbent is left to settle, and the supernatant is discarded; desorption with 500 µL of acetone and sonication for 5 min; separation of the sorbents and the supernatant is evaporated; the residue is redissolved in 100 μL of acetonitrile.	CE-UV	1.2 and 5 ng/mL	5–500 ng/mL	[53]
Lamivudine, Zidovudine, Efavirenz	500 µL of plasma	The sample is centrifuged and filtered, and 1 mL of water is added to 1 mL of plasma. PT-HM-MIP-SPE (poly(MAA-*co*-4-VP): 20 mg of the polymer is packed into a pipette tip (1000 mL, polypropylene); washing with 250 µL of water and sample load; washing with 300 µL of hexane; elution with 500 µL of methanol; the solution is evaporated, and the residue is redissolved in 50 µL of mobile phase.	HPLC-UV	n.a. and 0.25 μg/mL (Lamivudine, Efavirenz), n.a. and 0.05 μg/mL (Zidovudine)	0.25–10 μg/mL (Lamivudine, Efavirenz), 0.05–2 μg/mL (Zidovudine)	[54]
Aripiprazole, Dehydro-aripiprazole, Olanzapine, Risperidone, Paliperidone, Quetiapine, Clozapine	200 µL of plasma	μ-SPE (PRiME HLB (hydrophilic-lipophilic balance): 290 µL of 0.2% formic acid in water (pH 1.5) is added; sample load (2 × 255 µL) into the Oasis 96-well µElution Plate; washing with 400 μL (2 × 200 μL) of 5% methanol solution with water and 2% ammonia; vacuum is applied to dryness; elution with 200 μL (2 × 100 μL) of acetonitrile:methanol:buffer (formic acid, 0.2% at pH 3) solution (8:1:1, *v*:*v*:*v*); the eluate is collected in a 96-well plate, and 5 μL is injected into the chromatographic system.	LC-MS/MS (ESI)	n.a. and 0.18 ng/mL (Aripiprazole), n.a. and 0.25 ng/mL (Dehydro-aripiprazole), n.a. and 1 ng/mL (Olanzapine), n.a. and 0.70 ng/mL (Risperidone), n.a. and 0.20 ng/mL (Paliperidone), n.a. and 0.50 ng/mL (Quetiapine, Clozapine)	0.18–120 ng/mL (Aripiprazole), 0.25–80 ng/mL (Dehydro-aripiprazole), 1–100 ng/mL (Olanzapine), 0.70–60 ng/mL (Risperidone), 0.20–30 ng/mL (Paliperidone), 0.50–160 ng/mL (Quetiapine), 0.50–1000 ng/mL (Clozapine)	[55]
Amoxicillin, Penicillin, Tylosin tartrate, Roxithromycin, Clarithromycin, Azithromycin, Erythromycin, Chlorotetracycline hydrochloride, Terramycin, Tetracycline, Ofloxacin, Enrofloxacin, Ciprofloxacin, Norfloxacin, Olaquindox, Sulfamethazine, Sulfadiazine, Trimethoprim	500 µL of urine	QuEChERS DSPE: the sample is transferred to a polypropylene centrifuge tube, and 3 µL of formic acid is added; the mixture is vortexed, and 1 mL of methanol is added before shaking on a thermal shaker for 30 min at 20 °C; centrifugation for 10 min at 4 °C, and 1 mL of the upper layer is transferred to a roQ QuEChERS DSPE tube (200 mg); the solution is vortexed for 5 min, centrifuged for 10 min at 4 °C, and 200 μL of the final solution is transferred to a vial and stored at −20 °C for further analysis.	LC-MS/MS (ESI)	14.29 and 47.62 µg/L (Amoxicillin), 0.61 and 2.03 µg/L (Penicillin), 0.55 and 1.82 µg/L (Tylosin tartrate), 2 and 6.67 µg/L (Roxithromycin), 1.20 and 4 µg/L (Clarithromycin), 0.73 and 2.43 µg/L (Azithromycin), 31.43 and 104.76 µg/L (Erythromycin), 1.04 and 3.46 µg/L (Chlorotetracycline hydrochloride), 0.48 and 1.61 µg/L (Terramycin), 0.85 and 2.82 µg/L (Tetracycline), 1.79 and 5.98 µg/L (Ofloxacin), 1.42 and 4.72 µg/L (Enrofloxacin), 2.21 and 7.38 µg/L (Ciprofloxacin), 1.86 and 6.21 µg/L (Norfloxacin), 0.40 and 1.35 µg/L (Olaquindox), 0.37 and 1.23 µg/L (Sulfamethazine), 0.11 and 0.38 µg/L (Sulfadiazine), 0.14 and 0.45 µg/L (Trimethoprim)	0.34–1100 µg/L (Amoxicillin, Chlorotetracycline hydrochloride), 0.34–550 µg/L (Penicillin, Azithromycin, Erythromycin, Terramycin, Tetracycline, Ofloxacin, Enrofloxacin, Ciprofloxacin, Norfloxacin, Olaquindox, Trimethoprim), 0.34–275 µg/L (Tylosin tartrate, Roxithromycin, Clarithromycin, Sulfamethazine, Sulfadiazine)	[56]
Imatinib, Dasatinib, Nilotinib, Bosutinib, Ponatinib, Ruxolitinib, Ibrutinib, Filgotinib, Tofacitinib, Baricitinib, Peficitinib	300 µL of plasma	μ-SPE (PRiME MCX (Mixed-mode Cation exchange sorbent for bases)): 200 µL of 5% orthophosphoric acid in water is added; sample load (2 × 255 µL) into the Oasis 96-well µElution Plate; washing with 400 μL (2 × 200 μL) of 100 mM ammonium formate + 2% formic acid in aqueous solution and 400 μL (2 × 200 μL) of methanol; elution with 100 μL (2 × 50 μL) of 5% ammonium hydroxide in methanol solution (1:1, *v*:*v*) and 100 μL (1 × 100 μL) of water; after each step, vacuum is applied to dryness; the eluate is collected in a 96-well plate, and 5 μL is injected into the chromatographic system.	LC-MS/MS (ESI)	n.a. and 5 ng/mL (Imatinib), n.a. and 0.38 ng/mL (Dasatinib), n.a. and 4 ng/mL (Nilotinib), n.a. and 1 ng/mL (Bosutinib, Baricitinib), n.a. and 0.45 ng/mL (Ponatinib), n.a. and 1.50 ng/mL (Ruxolitinib, Tofacitinib), n.a. and 0.30 ng/mL (Ibrutinib), n.a. and 0.90 ng/mL (Filgotinib), n.a. and 2.50 ng/mL (Peficitinib)	5–5000 ng/mL (Imatinib), 0.38–400 ng/mL (Dasatinib), 4–4000 ng/mL (Nilotinib), 1–600 ng/mL (Bosutinib), 0.45–500 ng/mL (Ponatinib), 1.50–500 ng/mL (Ruxolitinib), 0.30–400 ng/mL (Ibrutinib), 0.90–1200 ng/mL (Filgotinib), 1.50–250 ng/mL (Tofacitinib), 1–250 ng/mL (Baricitinib), 2.50–900 ng/mL (Peficitinib)	[57]
Metoprolol	250 µL of plasma	MCC-DSPE: the sample is homogenized, and its pH is adjusted to 8; 10 µL of a zinc sulfate solution (0.05 mol/L) and 2.5 mg of MCC is added; agitation for 1 min, centrifugation for 4 min, and the supernatant is discarded; elution of the sorbent with 300 µL of methanol and agitation for 2 min; separation of the eluent by centrifugation for 5 min; evaporation and the residue is redissolved with 100 µL of the mobile phase.	HPLC-MS/MS (ESI)	0.30 and 0.5 ng/mL	1–1000 ng/mL	[58]

CE-UV: capillary electrophoresis-ultraviolet detection; D-µSPE: dispersive micro-solid-phase extraction; EI: electron ionization; ESI: electrospray ionization; GC-MS: gas chromatography-mass spectrometry; HPLC-MS/MS: high-performance liquid chromatography-tandem mass spectrometry; HPLC-UV: high-performance liquid chromatography-ultraviolet detection; LC-MS/MS: liquid chromatography-tandem mass spectrometry; LOD: limit of detection; LOQ: limit of quantitation; MCC-DSPE: microcrystalline cellulose-dispersive solid-phase extraction; n.a.: not available; Pipette tip SPE: pipette tip solid-phase extraction; PT-HM-MIP-SPE: pipette tip hollow mesoporous molecularly imprinted polymer solid-phase extraction; QuEChERS DSPE: quick, easy, cheap, effective rugged, safe-dispersive solid-phase extraction; μ-SPE: microelution-solid-phase extraction. In the absence of the LOQ value, the lowest point of the calibration curve was considered.

**Table 7 pharmaceutics-15-01055-t007:** Bioanalytical procedures using MSPE approaches for TDM.

Analytes	Sample (Amount)	Sample Pretreatment and Extraction Procedure	Analytical Technique	LOD; LOQ	Linear Range	Ref.
Venlafaxine, Paroxetine, Fluoxetine, Norfluoxetine, Sertraline	100 µL of plasma and 1 mL of urine	The plasma sample is incubated for 30 min and diluted to 1 mL with 5 mM PBS (pH 7); the urine sample is incubated for 30 min and diluted ten times with 5 mM PBS (pH 7.0), and the pH is adjusted with sodium hydroxide. MSPE (C_18_-Fe_3_O_4_@SiO_2_ NPs (functionalized magnetic silica nanoparticles)): 20 mg of the sorbent is preconditioned with 2 mL of methanol:water (1:1, *v*:*v*) by mechanical vibration for 15 min; the magnetic particles are gathered against the wall of vial by a magnet; the magnet is removed, and the particles are dispersed again; the supernatant is removed, and 1 mL of the previously prepared sample is added; extraction by mechanical vibration for 10 min; the sorbent is gathered against the inner wall of vail by a magnet, and the supernatant is directly poured and washed with 3 × 1 mL of water; elution with 200 µL of acetonitrile:0.1% formic acid (9:1, *v*:*v*) by mechanical vibration for 10 min and filtration.	UHPLC–MS/MS (ESI)	0.44 and 1.47 ng/mL (Venlafaxine), 0.75 and 2.46 ng/mL (Paroxetine), 0.52 and 1.7 ng/mL (Fluoxetine), 0.61 and 2.04 ng/mL (Norfluoxetine), 0.66 and 2.19 ng/mL (Sertraline) for plasma samples; 0.15 and 0.51 ng/mL (Venlafaxine), 0.40 and 1.34 ng/mL (Paroxetine), 0.21 and 0.70 ng/mL (Fluoxetine), 0.16 and 0.53 ng/mL (Norfluoxetine), 0.25 and 0.83 ng/mL (Sertraline) for urine samples	2.5–1000 ng/mL for all the compounds and both samples	[75]
Phenytoin sodium	100 µL of plasma	The sample is diluted with 300 µL of water (pH 5) and spiked with 50 µL of mobile phase. MSPE (Fe_3_O_4_@MIL-101(Cr)@MIP (molecularly imprinted polymers)): 8 mg of the sorbent is added into the solution and vortexed for 5 min; after absorption, the magnetic sorbents are separated using a magnet, and the supernatant is discarded; 500 µL of methanol is added and vortexed for 6 min; the elution solution is filtered.	HPLC-UV	n.a. and 0.05 µg/mL	0.05–40 µg/mL	[76]
Cyclophosphamide, Ifosfamide, Cisplatin, Methotrexate, Pemetrexed disodium, Capecitabine, 5-fluorouracil, Gemcitabine, Doxorubicin, Fulvestrant, Tamoxifen, Irinotecan	100 µL of plasma	MSPE: activation of the magnetic particles with 20 μL of hydrophilic–hydrophobic balance magnetic particles; 200 μL of methanol is transferred to a 96-well plate and stirred with a magnetic bar for 30 s; the activated magnetic particles are absorbed by the magnetic bar, transferred to another well plate and rinsed with 600 μL of water; 100 μL of the sample is added to another column of a 96-well plate and stirred with a magnetic bar for 30 s; elution of the drug-adsorbed magnetic particles that are absorbed by the magnetic bar are transferred to another well plate and rinsed with 600 μL of water for 30 s and then absorbed by the magnetic bar, transferred to a last column and rinsed with 800 μL of acetonitrile for 30 s to elute the analytes.	UPLC-MS/MS (ESI)	n.a. and 0.1 μg/mL (Cyclophosphamide, Ifosfamide, Cisplatin, Methotrexate, Pemetrexed disodium, Capecitabine, 5-fluorouracil, Gemcitabine), n.a. and 0.05 μg/mL (Doxorubicin, Fulvestrant, Tamoxifen, Irinotecan)	0.10–25 μg/mL (Cyclophosphamide, Ifosfamide, Cisplatin, Methotrexate, Pemetrexed disodium, Capecitabine, 5-fluorouracil, Gemcitabine), 0.05–12.5 μg/mL (Doxorubicin, Fulvestrant, Tamoxifen, Irinotecan)	[77]
Linezolid, Vancomycin, Teicoplanin, Tigecycline, Imipenem, Meropenem, Voriconazole, Micafungin	100 µL of plasma	MSPE: activation of the magnetic particles with 200 μL of methanol; 20 μL of the magnetic particles (0.1 g/mL) is introduced to a 96-well plate and stirred for 45 s; elution of methanol; the particles in the first step are transferred to another column of the 96-well plate by adsorption of the magnetic bar, and rinsed with 500 μL of water; 100 μL of the sample is added to another well plate and stirred for 45 s; elution of the drug-adsorbed magnetic particles that are transferred to another column of the 96-well plate by adsorption of the magnetic bar, rinsed with 500 μL of water and then transferred to a last column and rinsed with 600 μL of acetonitrile for 45 s to elute the analytes.	UPLC-MS/MS (ESI)	n.a. and 0.1 μg/mL (Linezolid, Teicoplanin, Tigecycline, Imipenem, Meropenem, Voriconazole Micafungin), n.a. and 0.2 μg/mL (Vancomycin)	0.1–25 μg/mL (Linezolid, Teicoplanin, Tigecycline, Imipenem, Meropenem, Voriconazole Micafungin), 0.2–50 μg/mL (Vancomycin)	[78]
Aletinib, Afatinib, Apatinib, Icotinib, Dasatinib, Erlotinib, Gefitinib, Crizotinib, Lapatinib, Regorafenib, Ceritinib, Sorafenib, Vemurafenib, Imatinib, N-desmethyl imatinib	100 µL of plasma	MSPE: activation of the magnetic particles with 150 μL of methanol; 20 μL of HLB magnetic particles is added to a 96-well plate and stirred with a magnetic bar for 30 s; the activated magnetic particles are absorbed by the magnetic bar and transferred to another column of the 96-well plate and rinsed with 600 μL of water; 100 μL of the sample is added to another well plate and stirred with a magnetic bar for 30 s; elution of the drug-adsorbed magnetic particles that are absorbed by the magnetic bar and transferred to another column of the 96-well plate, rinsed with 600 μL of water for 30 s and then absorbed by the magnetic bar and transferred to a last column and rinsed with 600 μL of acetonitrile for 30 s to elute the analytes.	UPLC-MS/MS (ESI)	n.a. and 2.5 ng/mL (Aletinib, Afatinib, Apatinib, Icotinib, Dasatinib, Crizotinib, Regorafenib, Vemurafenib, N-desmethyl imatinib), n.a. and 10 ng/mL (Erlotinib, Gefitinib, Lapatinib, Ceritinib, Sorafenib, Imatinib)	2.5–2500 ng/mL (Aletinib, Afatinib, Apatinib, Icotinib, Dasatinib, Crizotinib, Regorafenib, Vemurafenib, N-desmethyl imatinib), 10–10,000 ng/mL (Erlotinib, Gefitinib, Lapatinib, Ceritinib, Sorafenib, Imatinib)	[79]
Methotrexatein	50 µL of serum	MMIP-MSPE: 10 mL of methanol is added to the sample, the mixture is carried out by ultrasounds, centrifugation and collection of the supernatant; 10 mL of methanol is added to MMIP (100 mg) with stirring; after activation, the liquid is separated and discarded by magnetic separation, and the supernatant is added to 100 mg of MMIP; the sample is extracted and loaded by stirring at room temperature for 240 min; for magnetic separation and MMIP recovery, 5 mL of water:methanol (4:1, *v*:*v*) eluent is added with stirring; residual impurities are washed, and the material is recovered; 5 mL of the eluent methanol:acetic acid (4:1, *v*:*v*) is added to the treated MMIP with oscillation of the eluent for 60 min; after magnetic separation, the liquid is poured off, dried and redissolved in 500 µL of methanol solution and filtered.	HPLC-UV	12.51 and 50 ng/mL	50–250,000 ng/mL	[80]

ESI: electrospray ionization; HPLC-UV: high-performance liquid chromatography-ultraviolet detection; LOD: limit of detection; LOQ: limit of quantitation; MMIP-MSPE: magnetic molecularly imprinted polymer-magnetic molecularly imprinted solid-phase extraction; MSPE: magnetic solid-phase extraction; n.a.: not available; UHPLC-MS/MS: ultra-high performance liquid chromatography–tandem mass spectrometry; UPLC-MS/MS: ultra-performance liquid chromatography-tandem mass spectrometry. In the absence of the LOQ value, the lowest point of the calibration curve was considered.

**Table 8 pharmaceutics-15-01055-t008:** Advantages and disadvantages of the MSPE and SPDE procedures.

Extraction Technique	Advantages	Disadvantages
MSPE	- Simple and short duration; - Reduced solvent consumption; - Use of a single easily recoverable adsorbent; - Environmentally friendly; - Low cost.	- Low solubility in water and difficulty in recovery in dispersed medium in the case of nanotubes; - Graphene oxide-based adsorbents exhibit π–π stacking interactions between the graphene oxide nanosheets responsible for serious aggregation and re-stacking of the nanosheets, which results in a potential blockage of the active adsorption sites of the sorbent and a decrease in its specific surface. - Lack of multifunctional coatings.
SPDE	- Robustness of the capillary; - The device is not easily mechanically damaged.	- Carry-over; - The length of the coating may result in possible desorption problems.

**Table 9 pharmaceutics-15-01055-t009:** Advantages and disadvantages of the MIPs and TFME procedures.

Extraction Technique	Advantages	Disadvantages
MIPs	- Simple, stable, robust, flexible, selective and resistant to a wide range of pH, solvents and temperatures; - Extraction efficiency; - Emulates the interactions established by natural receptors to retain a target molecule; - Low-cost synthesis approach.	- Difficulties with optimization; - Length of time required for analysis; - Potential impossibility of long-term use due to analyte build-up.
TFME	- Enhanced sensitivity; - Fast mass transfer kinetics for both extraction and desorption processes; - Wide variety of available or potentially viable extractive phases; - Easy automation of the sampling step.	- Requires a larger volume of eluent during the desorption process; - Lack of conventional interfaces for on-line coupling of the device to the workflow of the analytical instrument; - The diffusion kinetics are slower in liquid phase, resulting in relatively longer desorption times; - Water can penetrate cellulose structure and destroy the network which may decrease the lifetime of the material and its reusability; - Modification of its surface brings an extra cost, and single use of such materials could make the analyses relatively costly.

**Table 10 pharmaceutics-15-01055-t010:** Bioanalytical procedures using MIP and TFME approaches for TDM.

Analytes	Sample (Amount)	Sample Pretreatment and Extraction Procedure	Analytical Technique	LOD; LOQ	Linear Range	Ref.
Quetiapine, Clozapine	200 μL of plasma and 500 μL of urine	The sample is diluted in 2 mL of water, and the pH value of the samples is adjusted to 9.5 and again diluted to volume with water for the next procedure. Magnetic ODS-PAN TFME (silica-coated magnetic nanoparticles (SiO_2_@Fe_3_O_4_)): a piece of thin film is preconditioned with methanol and water, added to the previously prepared sample, and the adsorption is performed by mechanical shaking for 50 min; the thin film is collected and rinsed with 3 mL of water; desorption is processed by mechanically shaking the magnetic thin film in 1 mL of methanol for 5 min; the obtained solution is evaporated, and the residue is redissolved in 100 μL of methanol.	HPLC-UV	0.013 and 0.05 μg/mL (Quetiapine), 0.015 and 0.054 μg/mL (Clozapine) for plasma samples; 0.003 and 0.01 μg/mL (Quetiapine, Clozapine) for urine samples	0.070–9 μg/mL (Quetiapine, Clozapine) for plasma samples; 0.012–9 μg/mL (Quetiapine, Clozapine) for urine samples	[91]
Sorafenib, Dasatinib, Erlotinib hydrochloride	2 mL of plasma, serum and n.a. of urine	100 μL of hydrochloric acid (12 mol/L) and 100 μL of trifluoracetic acid is mixed with 2 mL of plasma which is agitated, centrifuged and its supernatant separated and diluted with water (2:8, *v*:*v*), and the acid solution (pH 1) is neutralized with sodium hydroxide (0.01 mol/L) and filtered through a PVDF membrane. A total of 2 mL of acetonitrile is added to 2 mL of serum which is centrifuged and its supernatant separated, filtered and diluted with water (2:8, *v*:*v*) before extraction. The urine sample is centrifuged, filtered and diluted with water (5:5, *v*:*v*) before extraction. TF-SPME (polyfam/Co-MOF-74 composite nanofibers): the piece of sorbent (1 cm^2^) is cut from the nanofiber sheet and submerged in 10 mL of acetonitrile for 10 min for conditioning; it is immersed in 20 mL of the sample solution (optimum pH 10) for adsorption under agitation for 10 min for extraction; the sorbent is transferred to a vial to which 500 µL of alkaline methanol is added, with stirring for 7 min for the desorption process.	HPLC-UV	0.03 and 0.1 μg/L (Sorafenib), 0.15 and 0.5 μg/L (Dasatinib), 0.2 and 0.5 μg/L (Erlotinib hydrochloride) for all the samples	0.1–1500 μg/L (Sorafenib), 0.5–1500 μg/L (Dasatinib, Erlotinib hydro-chloride) for all the samples	[38]
Topiramate	n.a. of serum and 10 mL of urine	The serum is diluted 50 times in 0.1 M acetate buffer (pH 5); 100 µL of diluted serum sample is mixed with 5 mL of acetate buffer and transferred to an electrochemical cell. Urine is filtered and diluted in 3 mL of acetate buffer (pH 5). MIP/GO/GCE sensor and MIP/PVC/GCE sensor.	Voltammetry for MIP/GO/GCE sensor and Potentiometry for MIP/PVC/GCE sensor	5 × 10^−11^ and 2.7 × 10^−10^ mol/L (MIP/GO/GCE), 2.4 × 10^−10^ and 1 × 10^−9^ mol/L (MIP/PVC/GCE)	2.7 × 10^−10^–4.9 × 10^−3^ mol/L (MIP/GO/GCE), 1 × 10^−9^–3.4 × 10^−3^ mol/L (MIP/PVC/GCE)	[92]
Amitriptyline, Imipramine, Clomipramine, Desipramine, Doxepin, Trimipramine, Nortriptyline	700 μL of plasma	TF-MIP: the thin film is inserted into a vial containing 700 µL of plasma with 1% tri-ethylamine; the batch extraction process is carried out by agitation for 60 min; the thin film is washed by immersion in 1% aqueous triethylamine for 8 s; the thin film is dried and desorbed with 700 μL of 0.1% formic acid in 50% of aqueous acetonitrile for 20 min, then filtered.	UHPLC-MS/MS (ESI)	n.a. and 2.5 ng/mL (Amitriptyline), n.a. and 1 ng/mL (Imipramine, Clomipramine, Doxepin, Trimipramine, Nortriptyline), n.a. and 5 ng/mL (Desipramine)	2.5–500 ng/mL (Amitriptyline), 1–500 ng/mL (Imipramine, Clomipramine, Doxepin, Trimipramine, Nortriptyline), 5–500 ng/mL (Desipramine)	[93]
Favipiravir	10 μL of urine	The sample is mixed with 200 µL of acetonitrile and centrifuged; the supernatant is filtered and dried; the residue is dissolved and diluted with acetate buffer solution (pH 5) to a volume of 5 mL; 100 µL of the solution is added to an electrolytic cell for testing. MoS_2_@ MIP core-shell nanocomposite: the modified electrode is immersed into a blank acetate buffer solution (0.1 M, pH 5) for 5 min; the modified electrode is incubated with the sample for 5 min.	DPV	0.002 and 0.01 nM (3.14 × 10^−7^ and 1.57 × 10^−6^ μg/mL)	0.01–100 nM (1.57 × 10^−6^~1.57 × 10^−2^ μg/mL)	[94]
Methotrexatein	50 µL of serum	MMIP-MSPE: 10 mL of methanol is added to the sample, the mixture is carried out by ultrasound, centrifugation and collection of the supernatant; 10 mL of methanol is added to MMIP (100 mg) with stirring; after activation the liquid is separated, discarded by magnetic separation and the supernatant is added to 100 mg of MMIP; the sample is extracted and loaded by stirring at room temperature for 240 min; for magnetic separation and MMIP recovery, 5 mL of water:methanol (4:1, *v*:*v*) eluent is added with stirring; residual impurities are washed, and the material is recovered; 5 mL of the eluent methanol:acetic acid (4:1, *v*:*v*) is added to the treated MMIP with oscillation of the eluent for 60 min; after magnetic separation, the liquid is poured off, dried and redissolved in 500 µL of methanol solution and filtered.	HPLC-UV	12.51 and 50 ng/mL	50–250,000 ng/mL	[80]
Ceftazidime, Avibactam	n.a. of serum	Protein precipitation with three times the volume of methanol, centrifugation and collection of the supernatant which is diluted with the same volume of phosphate buffer saline (0.1 M, pH 7) for subsequent detection. MIP (N-Mo2C/SPE).	SWV	35 and 50 μM (Ceftazidime), 0.5 and 1 μM (Avibactam)	50–1000 μM (Ceftazidime), 1–1000 μM (Avibactam)	[95]
Imipenem, Piperacillin	n.a. of plasma and bronchoalveolar lavage (without validation)	TFME (96 DVB blades): sample extraction time is 30 min; as desorption solvent, a mixture of methanol:water (1:1, *v*:*v*) is used; the desorption time is 45 min.	LC-MS/MS (ESI)	n.a. and 0.01 mg/L for plasma samples	0.01–1 mg/L for plasma samples	[96]

DPV: differential pulse voltammetry; ESI: electrospray ionization; HPLC-UV: high-performance liquid chromatography-ultraviolet detection; LC-MS/MS: liquid chromatography-tandem mass spectrometry; LOD: limit of detection; LOQ: limit of quantitation; magnetic ODS-PAN TFME: magnetic octadecylsilane-polyacrylonitrile thin film microextraction; MIP/GO/GCE: molecularly imprinted polymer/graphene oxide/glassy carbon electrode; MIP/PVC/GCE: molecularly imprinted polymer/polyvinyl chloride/glassy carbon electrode; MMIP-MSPE: magnetic molecularly imprinted polymer-magnetic molecularly imprinted solid-phase extraction; MoS_2_@ MIP core-shell nanocomposite: flower-like molybdenum disulfide nanosphere and molecularly imprinted polymer; n.a.: not available; N-Mo_2_C/SPE: synthesized Mo_2_C with nitrogen doping/screen-printed electrode; SWV: square wave voltammetry; TFME: thin film microextraction; TF-MIP: thin film molecularly imprinted polymer; TF-SPME: thin film solid-phase microextraction; UHPLC-MS/MS: ultra-high performance liquid chromatography-tandem mass spectrometry. In the absence of the LOQ value, the lowest point of the calibration curve was considered.

**Table 11 pharmaceutics-15-01055-t011:** Advantages and disadvantages of the SBSE and FPSE procedures.

Extraction Technique	Advantages	Disadvantages
SBSE	- Simple, robust; - High sorption capacity, excellent extraction efficiency, high selectivity and sensitivity; - Low carry-over; - Sample volume and stirring speed greatly influence extraction efficiency - Possibility of automation and compatibility with different systems of analyte separation and detection.	- It requires a particular desorption unit; - Extraction time is longer; - Limited range of commercial coatings.
FPSE	- Simple, fast, low cost, highly efficient extraction and high chemical stability; - Small solvent consumption; - Fewer sample processing steps; - The fabric phase can be directly introduced into the sample and can be absorbed by the target substance; - Wide range of available materials; - Possibility of automation	- Extraction time is longer.

**Table 12 pharmaceutics-15-01055-t012:** Bioanalytical procedures using SBSE and FPSE approaches for TDM.

Analytes	Sample (Amount)	Sample Pretreatment and Extraction Procedure	Analytical Technique	LOD; LOQ	Linear Range	Ref.
Rifampicin	200 μL of plasma	SBSE (magnetic PDMS coated stir bar): the bar is conditioned for 24 h in a vial containing acetonitrile; plasma is placed in a vial to which 4 mL of 0.25 mol/L sodium acetate buffer (pH 5) is added; the vial is sealed, the stir bar is immersed in the sample, and the extraction is performed under magnetic stirring for 50 min; for desorption stir bar is removed, rinsed with water, dried and placed in a vial containing 1 mL of acetonitrile ensuring total immersion for magnetic agitation at 24 °C for 20 min; the stir bar is removed, the solvent is evaporated, and the residue is redissolved in 100 µL of mobile phase and 50 µL of hexane.	HPLC-UV	0.09 and 0.125 μg/mL	0.125–50 μg/mL	[107]
Fluoxetine, Sertraline, Citalopram, Paroxetine	800 μL of plasma	MEPS and SBSE (magnetic PDMS coated stir bar): the bar is conditioned for 24 h under stirring in a solution of acetonitrile:methanol (80:20, *v*:*v*); plasma is placed in a vial to which 4 mL of buffer solution is added; the vial is sealed, heated up to 50 °C, the stir bar is immersed into the sample, and the extraction is performed under magnetic stirring for 45 min; for desorption stir bar is removed, rinsed with water, dried and placed in a vial containing 1 mL of acetonitrile ensuring total immersion at 50 °C for 15 min; the stir bar is removed, the solvent is evaporated, and the residue is redissolved in 50 µL of acetonitrile.	NACE-DAD	n.a. and 20 ng/mL (Fluoxetine, Paroxetine), n.a. and 10 ng/mL (Sertraline), n.a. and 25 ng/mL (Citalopram) for SBSE technique	20–500 ng/mL (Fluoxetine, Paroxetine), 10–500 ng/mL (Sertraline), 25–500 ng/mL (Citalopram) for SBSE technique	[108]
Propranolol	n.a. of urine	The sample is diluted with water (1:4, *v*:*v*), the pH is adjusted to 9 by diluted ammonia, and 10 mL of the diluted sample is subjected to the extraction process. SBSE (GO/MIP coated stir bar): in a vial, 10 mL of sample solution is added with a GO/MIP coated stir bar, the vial is sealed and placed on a magnetic stir for 40 min; the stir bar is taken out of the sample solution, washed with water and dried; the stir bar is placed in a vial containing 100 μL of desorption solution of methanol and 10 mmol/L of sodium hydroxide (60:40, *v*:*v*), sealed and ultrasonicated for 20 min; the desorption solution is filtered.	HPLC-UV	0.37 and 1 μg/L	1–1000 μg/L	[109]
Ciprofloxacin, Sulfasalazine, Cortisone	180 μL of blood, 450 μL of plasma and 900 μL of urine	Blood sample is diluted with water (1:5, *v*:*v*). FPSE (sol-gel CW 20M (polyethylene glycol) coated): the membrane is cut into circular disks, cleaned with 2 mL of acetonitrile:methanol (50:50, *v*:*v*) for 5 min and rinsed 2/3 times in water; analytes extraction at TAAB rotator for 30 min; elution/back-extraction using 150 μL of methanol for 10 min and centrifugation.	HPLC-PDA	0.02 and 0.05 μg/mL for blood samples for all the compounds, 0.1 and 0.25 μg/mL for plasma samples for all the compounds, 0.03 and 0.10 μg/mL for urine samples for all the compounds	0.05–10 μg/mL for blood samples for all the compounds, 0.25–10 μg/mL for plasma samples for all the compounds, 0.10–10 μg/mL for urine samples for all the compounds	[110]
Fluoxetine	240 μL of plasma	SBSE (PDMS stir bar): the bar is conditioned for 24 h under magnetic stirring in a solution of methanol:acetonitrile (20:80, *v*:*v*); plasma is placed in a vial to which the SBSE bar and 3750 μL of sodium borate buffer (pH 9) is added, which is shaken; for the desorption step, 4000 μL of methanol:acetonitrile (75:25, *v*:*v*) are used, stirring for 50 min at a temperature of 50 °C; the bar is removed from the desorption solution, the sample is evaporated, and the residue is redissolved in 250 µL of desorption solution.	HPLC-FD	9.8 and 32.67 ng/mL	25–250 ng/mL	[111]
Febuxostat, Montelukast	20 μL of plasma	FPSE (sol-gel PCAP-PDMS-PCAP coated membrane): the membrane is cut into squares of 1 cm^2^ of surface, cleaned and activated by methanol:acetonitrile (40:60, *v*:*v*) for 5 min; it is rinsed in water, and 20 μL of plasma is diluted with 280 μL of isotonic solution (0.9% sodium chloride); the extraction is carried out under magnetic stirring at room temperature for 30 min; the back-extraction of the analytes from the membrane is carried out by means of 100 μL of methanol for 10 min.	HPLC-FLD	0.1 and 0.3 ng/mL (Febuxostat), 1.5 and 5 ng/mL (Montelukast)	0.3–10 ng/mL (Febuxostat), 5–100 ng/mL (Montelukast)	[112]
Favipiravir	350 μL of plasma and breast milk	FPSE (sol-gel PCAP-PDMS-PCAP coated membrane): the membrane is cut into squares of 1 cm^2^ of surface, cleaned and activated by methanol:water (60:40, *v*:*v*) for 5 min; it is rinsed in water and immersed into dilution of 350 μL of plasma and breast milk solutions, with 200 μL of serum physiologic (0.9% sodium hydroxide); the extraction is carried out under magnetic stirring at room temperature for 30 min; the back-extraction is carried out by using 500 μL of methanol for 30 min.	HPLC-UV	0.06 and 0.2 μg/mL for plasma samples, 0.15 and 0.5 μg/mL for breast milk samples	0.2–50 μg/mL for plasma samples, 0.5–25 μg/mL for breast milk samples	[113]

FPSE: fabric phase sorptive extraction; GO/MIP: water-compatible graphene oxides-molecularly imprinted polymers; HPLC-FD: high-performance liquid chromatography-fluorescence detection; HPLC-FLD: high-performance liquid chromatography-fluorimetric detection; HPLC-PDA: high-performance liquid chromatography-photo-diode array detection; HPLC-UV: high-performance liquid chromatography-ultraviolet detection; LOD: limit of detection; LOQ: limit of quantitation; MEPS: microextraction by packed sorbent; n.a.: not available; NACE-DAD: non-aqueous capillary electrophoresis-diode-array detection; PDMS: polydimethylsiloxane; SBSE: stir bar sorptive extraction; sol-gel PCAP-PDMS-PCAP: sol-gel poly (caprolactone)-blockpoly (dimethylsiloxane)-block-poly (caprolactone). In the absence of the LOQ value, the lowest point of the calibration curve was considered.

## Data Availability

Data is contained within the article.

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
