# Peer review of "Solid Phase-Based Microextraction Techniques in Therapeutic Drug Monitoring"

_pharmaceutics, 2023, doi:10.3390/pharmaceutics15041055_

Round 1
Reviewer 1 Report
This work is significant. It definitely falls within the scope of the Jounral Pharmaceutics. It is written in a clear and concise manner by subject matter experts. The writing and presentation are both excellent. My opinion as reviewer is that this work is on the verge of being accepted.
My only suggestion would be to include a methodological paragraph indicating the key words used in the selection of the articles presented here, as well as the criteria used to select these articles. The period, databases, and other details must be specified as well.
Author Response
Reviewer 1 (This reviewer’s suggestions are highlighted in green):
This work is significant. It definitely falls within the scope of the Journal Pharmaceutics. It is written in a clear and concise manner by subject matter experts. The writing and presentation are both excellent. My opinion as reviewer is that this work is on the verge of being accepted.
My only suggestion would be to include a methodological paragraph indicating the key words used in the selection of the articles presented here, as well as the criteria used to select these articles. The period, databases, and other details must be specified as well.
Response: Thank you for your suggestion. More information about the methodology used for the selection of articles was added to the manuscript.
Thank you for your positive comments and opinion! We truly appreciate.
Reviewer 2 Report
There are several points to be reviewed in the manuscript before it can be considered as fitted for publication. The English language also must be reviewed once the text is sometimes hard to follow.
Specific considerations:
Abstract: The great majority of the text of the abstract is dedicated to the definition and justification of TDM. The main theme of the manuscript, the solid-phase extraction techniques, is only mentioned in the last lines. Please review to include more information about the text of the article in the abstract.
Introduction:
For the definition of TDM, please refer to the website of the International Association of Therapeutic Drug Monitoring and Clinical Toxicology (IATDMCT).
Line 38: "In TDM, plasma samples are considered the gold standard, and this approach is routine practice for therapies involving immunosuppressants, antipsychotics, antiepileptics, antidepressants, antiarrhythmics, antivirals/antiretrovirals, anticancer, antibiotics and antifungal drugs [1,2,4]."
This is not completely correct once the most used immunosuppressants (cyclosporine and tacrolimus) are measured in whole blood for TDM purposes.
Line 67: "Since the rate of consumption of these classes of compounds is high, which can be considered a public health problem, it is important to compile the existing literature on the subject to assist health professionals in improving the quality of treatment for patients."
The phrase is difficult to understand. Why the consumption of these compounds is a problem? I understand that properly prescribed pharmacological treatments are intended to solve health problems. Please rephrase.
Rest of the text:
A "methods" section must be included with the description of the bibliographic search employed to obtain the data. The sources, constraints, and keywords must be presented, as well as the criteria to include a bibliography in the review. This method is presented in a very superficial way in the "sample pretreatment" section.
2.1. Microextraction by packed sorbent - Very superficial description. Describe the steps of a MEPS procedure, its advantages, and disadvantages.
Table 1, ref 17: "VAMS. Subsequently, for the solution resulting from the blood sample: evaporation, centrifugation, and reconstitution..." Describe the elution from VAMS.
Timeframe of reviews varies between reviewed techniques. Ex: 2017-2022 for MEPS, 2020-2022 for SPME, and so on. Why? Must be standardized.
Some information is repeated several times throughout the text, like the importance of drug measurements. Avoid these repetitions.
In the section "Park et al. [54] developed a methodology for the biomonitoring of 300 pesticides in 50 mg of hair samples. Using DSPE (PSA) as extraction technique and gas chromatography coupled to tandem mass spectrometry and LC-MS/MS, LOQ values between 2.5 and 7.5 pg/mg were obtained. The authors concluded that this multi
residue method is useful for biomonitoring pesticide exposure in situations such as operator risk assessments."
How pesticides in hair are within the scope of the manuscript?
Conclusions: Very generic, with several repetitions of the same terms, and presents objectives of the manuscript. Must be rewritten.
Author Response
Reviewer 2 (This reviewer’s suggestions are highlighted in yellow):
There are several points to be reviewed in the manuscript before it can be considered as fitted for publication. The English language also must be reviewed once the text is sometimes hard to follow.
Response: Thank you for your recommendation. English language was revised throughout the manuscript (marked with track changes).
Specific considerations:
Abstract: The great majority of the text of the abstract is dedicated to the definition and justification of TDM. The main theme of the manuscript, the solid-phase extraction techniques, is only mentioned in the last lines. Please review to include more information about the text of the article in the abstract.
Response: Thank you for your suggestion. The text of the abstract is limited to a maximum number of words, but it has been revised and more information on solid phase-based microextraction techniques was included.
Introduction:
For the definition of TDM, please refer to the website of the International Association of Therapeutic Drug Monitoring and Clinical Toxicology (IATDMCT).
Response: Thank you for your note. The International Association for Therapeutic Drug Monitoring and Clinical Toxicology website was referenced in the main manuscript.
Line 38: "In TDM, plasma samples are considered the gold standard, and this approach is routine practice for therapies involving immunosuppressants, antipsychotics, antiepileptics, antidepressants, antiarrhythmics, antivirals/antiretrovirals, anticancer, antibiotics and antifungal drugs [1,2,4]."
This is not completely correct once the most used immunosuppressants (cyclosporine and tacrolimus) are measured in whole blood for TDM purposes.
Response: We appreciate your observation. The sentence was modified in the main manuscript according to your opinion.
Line 67: "Since the rate of consumption of these classes of compounds is high, which can be considered a public health problem, it is important to compile the existing literature on the subject to assist health professionals in improving the quality of treatment for patients."
The phrase is difficult to understand. Why the consumption of these compounds is a problem? I understand that properly prescribed pharmacological treatments are intended to solve health problems. Please rephrase.
Response: Thank you for your recommendation. The sentence was rephrased in the main manuscript.
Rest of the text:
A "methods" section must be included with the description of the bibliographic search employed to obtain the data. The sources, constraints, and keywords must be presented, as well as the criteria to include a bibliography in the review. This method is presented in a very superficial way in the "sample pretreatment" section.
Response: Thank you for your suggestion. As requested, a methods section was added to the manuscript.
2.1. Microextraction by packed sorbent - Very superficial description. Describe the steps of a MEPS procedure, its advantages, and disadvantages.
Response: Thank you for your recommendation. The steps of the MEPS extraction procedure were described in the main manuscript text and the advantages and disadvantages were added in a Table.
Table 1, ref 17: "VAMS. Subsequently, for the solution resulting from the blood sample: evaporation, centrifugation, and reconstitution..." Describe the elution from VAMS.
Response: Thank you for your note. The information about VAMS extraction was added to Table 1.
Timeframe of reviews varies between reviewed techniques. Ex: 2017-2022 for MEPS, 2020-2022 for SPME, and so on. Why? Must be standardized.
Response: Thank you for your question. More information about the timeframe used for the literature research was added to the manuscript. The range of years of research for this review varies between the different techniques according to the number of articles published, in order to present a reasonable number of papers for each technique, emphasizing those most implemented in recent years.
Some information is repeated several times throughout the text, like the importance of drug measurements. Avoid these repetitions.
Response: Thank you for your recommendation. The manuscript was revised and repetitions were corrected.
In the section "Park et al. [54] developed a methodology for the biomonitoring of 300 pesticides in 50 mg of hair samples. Using DSPE (PSA) as extraction technique and gas chromatography coupled to tandem mass spectrometry and LC-MS/MS, LOQ values between 2.5 and 7.5 pg/mg were obtained. The authors concluded that this multi residue method is useful for biomonitoring pesticide exposure in situations such as operator risk assessments."
How pesticides in hair are within the scope of the manuscript?
Response: Thank you for your question. The article on determination of pesticides in hair samples was added to this review in order to present other applications of the dispersive solid-phase extraction technique for biomonitoring purposes. To avoid misleading the reader, the sentence about this work was removed from the main manuscript.
Conclusions: Very generic, with several repetitions of the same terms, and presents objectives of the manuscript. Must be rewritten.
Response: Thank you for your recommendation. The manuscript was revised and some parts were rewritten.
Thank you for your comments and suggestions!
Reviewer 3 Report
The review is comprehensive and well written, It can be published. Inclusion of some figure or scheme illustrating the solid phase micro-extraction technique would add value to the manuscript and help the reader. Also the authors should provide some information on the recognition forces involved in solid-phase extraction.
Author Response
Reviewer 3 (This reviewer’s suggestions are highlighted in pink):
The review is comprehensive and well written, It can be published. Inclusion of some figure or scheme illustrating the solid phase micro-extraction technique would add value to the manuscript and help the reader. Also the authors should provide some information on the recognition forces involved in solid-phase extraction.
Response: Thank you for your suggestions. Some representative figures of solid phase-based microextraction techniques were added to the manuscript.
In addition, information on the extraction mechanisms involved in solid-phase extraction was also provided.
Thank you for your opinion! We truly appreciate.
Reviewer 4 Report
In this manuscript a review is presented that compiles solid-phase microextraction based methods aimed at therapeutic drug monitoring (TDM). The paper may help the readers of Pharmaceutics to get an overview of the methods developed for TDM of different classes of drugs that involve microextraction, including experimental conditions used and some relevant analytical parameters. However, in my opinion, different changes are needed before acceptance. The major points to be considered are:
- The criteria used for selecting the years of the literature research done are not clear. Even more, the time interval studied is highly variable depending on the microextraction technique considered. As an example, in lines 166 and 167 it is stated that table 2 lists methods based on SPME (fiber SPME, I guess) between years 2020 and 2022, and in-tube SPEM in the period 2010-2022; lines 248-249: between years 2010-2022 for pipette tip SPE, and between 2017-2022 for DSPE. This is somewhat confusing. Besides, the information concerning the relative use (relative number of applications for the same time period) is lost. I recommend the authors to set a specific time period, the same for all microextraction techniques, so that the reader can get a better picture of the relative use of the different approaches in the context of TDM.
- The information given in the text is often redundant, as most part of it is also shown in the preceding tables. Thus, in some parts of the manuscript the text adds limited value. My recommendation is, for each of the techniques considered, to condense/remove those parts of the text to avoid repetition with respect the tables. Instead, a more elaborated comparison of the different proposals should be added. The comparison could include for each microextraction technique topics such as the types of drugs more used for, resources and time of analysis required for the sample treatment, availability of the extraction devices used, analytical capabilities of methods developed…. Without this comparison, the manuscript is a mere compilation of articles.
- (Related to the above point) A more elaborated discussion about the different techniques included in the work is needed. In fact, the Conclusions and future perspectives section only presents general well know information about microextraction. On the basis of the information gathered by the authors, at least the following topics need to be addressed: are there solid phase microextraction techniques better suited for TDM than the others?, or better suited for a specific type of drugs?; are there techniques that stand out from the others in the very last few years? or which are gradually being replaced by others?; any others trends detected? In addition, despite the tittle of the section, discussion about future perspectives in the field is missing. This section must be substantially improved.
Minor points:
- Line 139: methods instead of methodologies is more appropriate.
- Line 159: inner is not correct here; and there are multiple coatings for in-tube SPME
- Please, clarify why some of the references describing SPME-based methods are not included in the corresponding tables (example, refs 53-55 in lines 283-296).
- I guess that the references should be cited by chronological order? (example, line 247, refs. 44 and 45).
Author Response
Reviewer 4 (This reviewer’s suggestions are highlighted in gray):
In this manuscript a review is presented that compiles solid-phase microextraction based methods aimed at therapeutic drug monitoring (TDM). The paper may help the readers of Pharmaceutics to get an overview of the methods developed for TDM of different classes of drugs that involve microextraction, including experimental conditions used and some relevant analytical parameters. However, in my opinion, different changes are needed before acceptance. The major points to be considered are:
The criteria used for selecting the years of the literature research done are not clear. Even more, the time interval studied is highly variable depending on the microextraction technique considered. As an example, in lines 166 and 167 it is stated that table 2 lists methods based on SPME (fiber SPME, I guess) between years 2020 and 2022, and in-tube SPME in the period 2010-2022; lines 248-249: between years 2010-2022 for pipette tip SPE, and between 2017-2022 for DSPE. This is somewhat confusing. Besides, the information concerning the relative use (relative number of applications for the same time period) is lost. I recommend the authors to set a specific time period, the same for all microextraction techniques, so that the reader can get a better picture of the relative use of the different approaches in the context of TDM.
Response: Thank you for your recommendation. More information about the criteria used for the literature research was added to the manuscript. The time period of research for this review varies between the different techniques according to the number of articles published, in order to present a reasonable number of works for each technique, emphasizing those most implemented in recent years.
The information given in the text is often redundant, as most part of it is also shown in the preceding tables. Thus, in some parts of the manuscript the text adds limited value. My recommendation is, for each of the techniques considered, to condense/remove those parts of the text to avoid repetition with respect the tables. Instead, a more elaborated comparison of the different proposals should be added. The comparison could include for each microextraction technique topics such as the types of drugs more used for, resources and time of analysis required for the sample treatment, availability of the extraction devices used, analytical capabilities of methods developed…. Without this comparison, the manuscript is a mere compilation of articles.
Response: Thank you for your suggestion. The manuscript was revised and repetitions were corrected. In order to overcome the unfeasibility of carrying out a direct comparison between the works of the different techniques, since they determine different classes of compounds, or analyse different biological samples in different volumes and do not allow a correlation between the different extraction steps for each of the techniques, several Tables were also added in order to compare the listed topics.
(Related to the above point) A more elaborated discussion about the different techniques included in the work is needed. In fact, the Conclusions and future perspectives section only presents general well know information about microextraction. On the basis of the information gathered by the authors, at least the following topics need to be addressed: are there solid phase microextraction techniques better suited for TDM than the others?, or better suited for a specific type of drugs?; are there techniques that stand out from the others in the very last few years? or which are gradually being replaced by others?; any others trends detected? In addition, despite the tittle of the section, discussion about future perspectives in the field is missing. This section must be substantially improved.
Response: Thank you for your recommendation. The conclusions and future perspectives section was revised and improved according to the suggested topics.
Minor points:
Line 139: methods instead of methodologies is more appropriate.
Response: Thank you for your note. The term was replaced as suggested.
Line 159: inner is not correct here; and there are multiple coatings for in-tube SPME
Response: Thank you for your suggestion, this was modified.
Please, clarify why some of the references describing SPME-based methods are not included in the corresponding tables (example, refs 53-55 in lines 283-296).
Response: Thank you for your question. Some references of works, not included in the tables, that describe methods based on these microextraction techniques, were cited with the intention of presenting other possible applications of published articles. These papers were considered for text only because they do not meet all the criteria defined for inclusion in the tables, do not identify the topics included in the tables, or because they were only applied for the purpose of biomonitoring compounds that do not have narrow therapeutic windows.
I guess that the references should be cited by chronological order? (example, line 247, refs. 44 and 45).
Response: Thank you for your note. The references were made using the Mendeley software, which manages the order in which they are placed. However, where possible, references were placed in chronological order throughout the manuscript.
Thank you for your comments and suggestions!
Reviewer 5 Report
1. Tables 3 and 4 should be made concise.
2. Incorporate a figure describing the microextraction technique.
3. Mention the pharmaceutical relevance of TDM in the manuscript.
Author Response
Reviewer 5 (This reviewer’s suggestions are highlighted in blue):
- Tables 3 and 4 should be made concise.
Response: We appreciate your observation. Tables 3 and 4 were modified as suggested.
- Incorporate a figure describing the microextraction technique.
Response: Thank you for your suggestion. Some representative figures of solid phase-based microextraction techniques included in this review were added to the manuscript.
- Mention the pharmaceutical relevance of TDM in the manuscript.
Response: More information regarding the relevance of TDM was added to the manuscript.
Thank you for your comments and suggestions!
Reviewer 6 Report
The authors perform a good systematic literature review on sample preparation techniques based on solid-phase microextraction for drug detection in therapeutic monitoring situations. The papers reviewed are relevant and the information is well analyzed.
I recommend accepting the manuscript after two minor corrections.
1. In the abstract, some results should be presented in addition to some general conclusions.
2. The conclusions and future perspectives are very brief, they do not really represent the excellent review that the authors performed. It should be explicit, it really presents conclusions, not comments.
Author Response
Reviewer 6 (This reviewer’s suggestions are highlighted in dark green):
Comments and Suggestions for Authors
The authors perform a good systematic literature review on sample preparation techniques based on solid-phase microextraction for drug detection in therapeutic monitoring situations. The papers reviewed are relevant and the information is well analyzed.
I recommend accepting the manuscript after two minor corrections.
- In the abstract, some results should be presented in addition to some general conclusions.
Response: The abstract was completed with more information, as requested.
- The conclusions and future perspectives are very brief, they do not really represent the excellent review that the authors performed. It should be explicit, it really presents conclusions, not comments.
Response: Thank you for your positive evaluation. More information regarding the conclusions and future perspectives was added to the manuscript.
Thank you for your comments and suggestions!
Round 2
Reviewer 2 Report
The comments of this reviewer were adequately addressed.
Reviewer 4 Report
After the modifications made, I think the paper is suitable for publication.